# Extracellular Matrix Cues Regulate Mechanosensing and Mechanotransduction of Cancer Cells

**DOI:** 10.3390/cells13010096

**Published:** 2024-01-02

**Authors:** Claudia Tanja Mierke

**Affiliations:** Biological Physics Division, Peter Debye Institute of Soft Matter Physics, Faculty of Physics and Earth Science, Leipzig University, Linnéstraße 5, 04103 Leipzig, Germany; claudia.mierke@uni-leipzig.de

**Keywords:** extracellular matrix (ECM) remodeling, fibronectin, cell mechanics, cancer cells, EMT, immune cells, plasticity, mechanosensing, cancer-associated fibroblasts (CAFs)

## Abstract

Extracellular biophysical properties have particular implications for a wide spectrum of cellular behaviors and functions, including growth, motility, differentiation, apoptosis, gene expression, cell–matrix and cell–cell adhesion, and signal transduction including mechanotransduction. Cells not only react to unambiguously mechanical cues from the extracellular matrix (ECM), but can occasionally manipulate the mechanical features of the matrix in parallel with biological characteristics, thus interfering with downstream matrix-based cues in both physiological and pathological processes. Bidirectional interactions between cells and (bio)materials in vitro can alter cell phenotype and mechanotransduction, as well as ECM structure, intentionally or unintentionally. Interactions between cell and matrix mechanics in vivo are of particular importance in a variety of diseases, including primarily cancer. Stiffness values between normal and cancerous tissue can range between 500 Pa (soft) and 48 kPa (stiff), respectively. Even the shear flow can increase from 0.1–1 dyn/cm^2^ (normal tissue) to 1–10 dyn/cm^2^ (cancerous tissue). There are currently many new areas of activity in tumor research on various biological length scales, which are highlighted in this review. Moreover, the complexity of interactions between ECM and cancer cells is reduced to common features of different tumors and the characteristics are highlighted to identify the main pathways of interaction. This all contributes to the standardization of mechanotransduction models and approaches, which, ultimately, increases the understanding of the complex interaction. Finally, both the in vitro and in vivo effects of this mechanics–biology pairing have key insights and implications for clinical practice in tumor treatment and, consequently, clinical translation.

## 1. Introduction

The extracellular matrix (ECM) encompasses an intricate, dynamic, and crosslinked reticulation that contains tethered biomolecules [1,2]. The proper function of tissues and entire organs relies on the function of the ECM scaffold. It provides vital physical sustenance to cells and produces key biochemical and biomechanical cues that are necessary for the development of tissues. The ECM is generated and remodeled by dynamic, reciprocal, biochemical, and biophysical interactions between the ECM and various cells, such as fibroblasts [3], cancer-associated fibroblasts (CAFs) [4], adipocytes [5], cancer-associated adipocytes [6], cancer-associated macrophages [7], cancer cells [8], and stem cells [9,10]. These interferences between cells and the ECM play a role in numerous physiological and pathological processes, involving homeostasis, aging, wound healing, and multiple diseases, like cancer, fibrosis, and cardiovascular and pulmonary pathologies [11,12,13,14,15]. Altering the interactions between the ECM and cells may help to regulate cell behavior, offering great potential for future treatment options. More specific, knowledge of the interactions between the ECM and cancer stem cells (CSCs) will be beneficial in finding advanced and effective therapeutic approaches to eradicate CSCs [16].

In cell biology and physiology science, the idea that physical characteristics affect biological structure and function has a long-standing tradition. The interaction between the ECM and the cells underlie dynamical adaptation and relies on forces [17]. Notably, the ECM is continuously restructured by forces either intrinsic to the cell or external to it, rendering it extremely dynamic in nature. Apart from the restructuring of the ECM, there exists a large compositional heterogeneity. Depending on the organ type, the ECM must have specific biochemical and mechanical properties, such as tensile and compressive strength, topology, and elasticity. Cells encounter extrinsic mechanical inputs such as shear, tensile, and compressive forces and determine cellular responses to sustain the tissue’s structural integrity and operability [18]. Cells perceive their environment via membrane receptors, such as integrins and cadherins [19]. Upon application of a mechanical load to adhesion receptors, force-induced functionalities are enabled, such as alterations in protein conformation or modifications in enzyme-catalyzed reactions. Among them are the unfolding of talin [20,21,22,23], the activation of focal adhesion kinase (FAK) [24], and the unfolding and activation of vinculin [25,26,27,28]) that, in return, trigger biochemical signaling, which is referred to as mechanosignaling, such mitohornesis (a biochemical process in which the activation of mitochondria leads to an increase in free radicals in the cell, which finally causes an activation of the cell’s own defenses against oxygen radicals) [29]. These biomechanically triggered biochemical signals promote successive cellular reactions, such as polarity, migration, differentiation, and survival, to accommodate physiological cues [30]. Understanding the mechanical crosstalk and signal transduction interface of cells and ECM mimetics is a key enabling strategy in the identification of cell–ECM specific interactions. Consequently, it can be stated that the field of mechanobiology, which relates bidirectional and dynamic interactions at the mechanical and biological levels, is of increasing interest to many cell biologists and biophysicists because genetics and biochemistry alone are not sufficient to adequately clarify biological form and function. The microenvironment enveloping cells in vivo and in vitro can serve a huge part in controlling cell performances. Therefore, the mechanical facets of this scenery, which is referred to as the mechanoscape, are equally critical for developing both an understanding of cell behavior and tools to imitate it. The majority of adherent cell types can actively perceive the mechanical features of their environment by imposing a contractile force that is imparted to cell–matrix or cell–cell adhesions. Among the mechanical characteristics are passive mechanical aspects of the ECM that encompass bulk and local stiffness and viscoelasticity, density of ligands, and topography [31].

Cells generate components of the ECM and can alter its structural and mechanical organization. Thereby, cells are able to modify largely the ECM composition and cell–matrix adhesion features, all of which addresses the mechanical characteristics of the ECM. These mechanical cue alterations of the ECM represent a direct outcome of cellular activity, which, consequently, establishes a principle of dynamic reciprocity between the cell and the enveloping microenvironment [32,33]. Reversely, cells can passively receive mechanical signals when the ECM applies a force to them when tissues are deformed by shear, stretch, or compression, which is aided by static or cyclic mechanical loading [34]. Cells can also communicate with one another over wide distances through traction-induced displacements of the ECM fiber scaffold.

A major development in this field have been the design of ECM-mimicking biomaterials with different biophysical or biochemical characteristics. Thereby the biomaterials have been built with specific mechanical cues, including stiffness, viscosity, degradability, and diffusivity, all of which can be regulated with high precision. Thus, the cellular reaction of one mechanical cue or combined mechanical cues can be explored [35,36]. Even more advanced smart biomaterials with dynamic characteristics have been invented that mirror the microenvironment more accurately, that can help to understand the mechanisms regulating the mechanoresponse of cells [37,38,39]. In this review, passive and active characteristics of the ECM are presented. It is discussed how cells perceive and react to the individual mechanical cues of the ECM of solid tumors. The effect of the ECM on cellular mechanics and biochemical cues on various length scales is discussed. Thereby, the frontiers of the field of mechanotransduction are determined and future directions, such as therapy improvements, are highlighted. In addition, the improvement of manipulation of biophysical techniques is at the forefront of research and numerous techniques are close to clinical translation. Finally, a complex picture of the interplay between different static and dynamic mechanical factors of the ECM at different levels is created. In this way, the control of cellular mechanosensors and functions becomes possible.

## 2. Mechanotransduction Mechanism: Cell–Matrix Force Relationships with Emphasis on Matrix or Tissue Forces

First of all, there is a specific mechanical cue within a tumor, the extracellular fluid (ECF) viscosity. The ECF exhibits a distinctive density, viscosity, and osmotic pressure. Precursors and decomposition substances of biomaterials increase the compaction (crowding) of the ECF and frequently enhance its viscosity. In addition, a rise in ECF viscosity is associated with mucin-producing adenocarcinomas; viscosity-increasing polymer solutions encourage the mesenchymal-like cell migration of liver cancer cell lines [40]. The viscosity of mucus is broadly dependent on mucin concentration, temperature, pH, ionic strength, and shear rate. Viscosity increases the integrin-dependent propagation velocity of cells and induces a restructuring of the actin cytoskeleton, resulting in an enlargement of the cell area, a flattening of the nucleus and a relocation of YAP and β-catenin, proteins participating in mechanotransduction. Apart from the ECF viscosity, there are cell–matrix force interferences that generate matrix and tissue forces.

Besides biological and physical examination, cell–ECM interactions in the cancer microenvironment hold potential for clinical studies, mainly due to the fact that the somatic mutation rate tends to be in proportion to the stiffness of normal tissue [41] (Figure 1). Drugs targeting the ECM seek to suppress specific matrix interactions that lead to ECM-driven chemoresistance or to modify the tumor microenvironment to improve the regulation of cell function or drug distribution. ECM generation within the tumor is strongly increased, which, in the majority of instances, causes increased stiffness in comparison to healthy tissue [42]. This increased matrix stiffness, which seems to coincide with more tightly clustered ECM fibers, poses two problems: Firstly, the enhanced stiffness may encourage the metastatic activity of cancer cells [43], and, secondly, the transport of medications, and possibly immune cells, across the tumor is impeded [44]. Inhibitory substances for TGF-β, for instance, decrease the release of ECM proteins [45] to circumvent additional ECM modification. Since the process of metastasis of cancer cells is the main reason for death, a number of medications have been designed to impede the migration of metastatic cells [46]. These metastatic cells propel their path through the human body via the breakdown of the ECM through the generation of matrix metalloproteinases (MMPs) or other matrix-reducing enzymes, such as heparanase. Numerous treatments are targeted at blocking the formation of such enzymes to impede the invasion of invasive cells [47,48]. Immuno-oncology [49] represents a fast-evolving field of translation with the advent of new and clinically effective molecules, for example, checkpoint inhibitors, and genetically engineered cells, for instance, cancer antigen receptor T cells or CAR-Ts. T-cell activation has been found to be susceptible to nanoscale antigen distances in the ECM [50], consistent with fundamental mechanobiological findings on ligand display. Nevertheless, the effective use of these reagents targeting solid tumors [51] is still difficult to imagine and generally depends on the penetration of immune cells hitting the physical barriers discussed earlier.

### 2.1. Tensile Force

Primary tumors are enveloped by an ECM, in which the most abundant protein is type I collagen. The malignant advancement of the tumor is coupled to specific collagen structural arrangements [52], referred to as tumor-associated collagen signatures (TACSs). TACSs have been linked to the patient’s prognosis. The parallel collagen organization seen at the tumor boundary and the radial alignment in the invasion zone has prompted the question of the mechanisms that govern the organization of these structures. The impact of contractile forces that originated from tumor spheroids incorporated in a biomimetic collagen I matrix have been explored. It has been found that contractile forces act directly after the sowing and distort the ECM, which results in radial tensile forces inside the matrix. First of all, there is an accumulation of collagen in the adjacent tissue (TACS-1). In the later phases, the collagen fibers orient themselves parallel to the surface of the tumor (TACS-2) [52,53,54]. Lastly, in invasive tumors, the collagen fibers are oriented perpendicular to the tumor border (TACS-3), which also relates to the general direction of cellular invasion [55]. TACSs have been reported to be a prognostic indicator for the survival of patients [53]. In a similar vein, a powerful relationship between metastatic potency and the orientation of the intra-tumoral matrix, comprising the radial and parallel orientation of collagen fibers, has been demonstrated in a mouse model for colorectal carcinoma [56]. Positive feedback between the tumor-driven alterations in the collagen and the cancer and cancer-associated cell types has been hypothesized [57], which could account for the robust and consistent emergence of these collagen patterns.

The relaxing of this tension by cutting the collagen decreases invasion, demonstrating a mechanical link between the tensile state of the ECM and invasion. These results again indicate that tensile forces in the ECM ease invasion. In addition, the simultaneous ECM contraction and growth of the tumor causes the collagen to condense and realign on the surface of the spheroid. A tension-based model has been used to clarify collagen organization and the initiation of invasion due to tumor-derived forces, which is termed tension-driven invasion mode.

These observations can be grouped into a model of invasion that is rooted in the self-restructuring of the 3D matrix through the spheroid of CT26 cancer cells (Figure 2) [58]. Following the sowing of the spheroid in the collagen that is distributed in a random manner, the first cell layer forms a joint to the fibers with the assistance of adhesion proteins. They perceive the stiffness, tension, and fiber orientation, and initiate certain biochemical cues that further reshape the matrix. In addition to this biochemical restructuring, the cells begin to generate tensile forces and pull the collagen together in a radial pattern surrounding the spheroid (Figure 2, black arrows). As a consequence of this radial contraction, the fibers on the surface become aligned in a parallel direction to the spheroid, whereas the fibers in the far distal sections become radially aligned and are subjected to tension (Figure 2, lilac arrows). This is illustrated in blue in the simplified geometry of the ECM at the top of the sketch. The expanding spheroid displaces the collagen fibers onto the surface, which results in a subsequent rise in collagen density and in parallel alignment in the first collagen layer (Figure 2, blue arrows).

In addition to these rather steric motives for the parallel collagen alignment, both biochemical restructuring and forced repositioning as a result of tangential forces are potential reasons. The cells create radial protrusions that contract the exterior collagen, thereby enhancing the mechanical tension exerted by the cells and transferred to their neighboring cells. As soon as critical tension is achieved, the cell–cell contract breaks and the cells can escape the spheroid. The invading cells then orient themselves along the radial fibers and move away from the spheroid. Cells from the subjacent layers join in and exert further traction, leading to an intensification of the tension, which, ultimately, causes the collagen to stiffen and the contraction to diminish. When collagen fibers are cleaved and a vacant surface is generated, the cells in the direction of the cut are unable to produce sufficient tension in the ECM to disengage from the spheroid. In addition, force-dependent intracellular cues may not be elicited in this case. In this model, the alignment of the fibers is accounted for as a straightforward geometric characteristic of the expansion of the spheroids and the contraction of the collagen. The simultaneous initiation of invasion is, therefore, the consequence of a force equilibrium that forecasts a stress balance across the spheroid and a reasonable critical stress governed by the cell-to-cell adhesion strength. Radial outgrowth can be attributed to the anisotropic tension and the radial orientation of the fibers. After all, the reduced invasion with diminishing tension represents a direct forecast of the model.

### 2.2. Hydrostatic Pressure

Pulmonary hypertension (PH) denotes a pathophysiological situation with elevated blood pressure in the pulmonary arteries, which may be caused by a number of factors [59], such as cancer and its malignant progression [60]. A hallmark of PH is the alteration of small pulmonary arteries resulting from hyperproliferation of vascular smooth muscle cells (VSMCs) and myofibroblasts, causing the aberrant accumulation of collagen and elastin and the stiffening of the vascular ECM. A series of cross-sectional investigations indicate that YAP/TAZ activation, which is subsequent to ECM stiffening, appears to be a major factor in PH [61]. Vascular restructuring and stiffening activate YAP/TAZ, which then controls a transcriptional program that enhances ECM laydown and crosslinking and reinforces vascular restructuring and stiffening, resulting in a feedforward feedback cycle. In particular, the stiffening of the ECM activates YAP/TAZ in myofibroblasts, endothelial cells, and vascular smooth muscle cells (VSMCs), and thereby promotes the multiplication of these cells. Moreover, YAP/TAZ within these cells lead to the activation of genes that contribute to the production of ECM, such as collagens, and crosslinking, such as LOX [61]. Moreover, YAP/TAZ provide a connection between mechanical irritation and the disordered vascular metabolism related to PH. ECM remodeling regulates glutaminase expression through the activation of YAP/TAZ, which results in the activation of glutaminolysis and anaplerosis and favors the proliferation and migration of pulmonary artery endothelial cells (PAECs) and pulmonary artery smooth muscle cells (PASMCs). LOX inhibitors attenuated nuclear YAP/TAZ and ameliorated terminal PH symptoms in mouse models [62]. Increased pulsatility and shear stress have been related to YAP/TAZ activation in reshaping pulmonary vascular ECM and the proliferation of pulmonary adventitial myofibroblasts. However, it is not clear whether mechanical cues from elevated pulsatility and shear stress by itself are adequate to activate YAP/TAZ in adventitial myofibroblasts in the lack of a rigid matrix [63].

What is the impact of high hydrostatic pressure (HHP) on cells? HHP constitutes a physical factor that influences cell physiology. Inadequate pressure can inhibit cell growth, lead to structural damage to cells, and cause cell death. HHP between 1 and 100 MPa is regarded as being non-lethal, resulting in reversible morphological alterations and a mild stress reaction. HHP between 100 and 150 MPa can trigger apoptosis of mouse cells. HHP between 150 and 250 MPa can impair the vitality of human cells, whereas pressures between 300 and 400 MPa can result in the necrosis of cells [64,65,66,67]. The pressure is immediately and evenly dispersed over the entire non-toxic medium during inactivation using the HHP treatment and can be transferred across all flexible substrates. The treatment is applied to every part of the specimen at the same time with the identical pressure. Ultimately, every single modified cell in the system is exposed to precisely the identical load, so that it is possible to attain extremely high levels of consistency [68]. The pressure enters the cell instantly and entirely, affecting all intracellular constituents [68]. It is hypothesized that the pressure to which cells are subjected, beyond a specific limit, leads to a continuous rise in membrane stiffness and protein denaturation, which finally results in cell death [69].

Some studies have found evidence of both the apoptosis and necrosis of cells following non-physiological exposure to HHP, but the specific mechanism of cell death is largely a function of cell-type sensitivity and magnitude of pressure [67,70]. Cell death due to apoptosis takes place at a pressure of about 200 MPa [71], and cell necrosis develops at a pressure of more than 300 MPa [64,71,72]. HHP has been characterized to inactivate B16-F10 melanoma cells at various pressures (≥50 MPa) and for several time periods (≥1 min) [73]. Their findings indicate that HHP can be an efficacious melanoma vaccine generation tool when the pressure is ≥200 MPa and the duration of treatment is ≥30 min. It has been proven that in vitro treatment at 200 MPa or higher fully blocks the development of cancer cell colonies and that HHP generates inactivated cancer cells that can be employed as a tumor vaccine [74]. Similarly, there seems to be a synergy between cancer-cell-derived vaccines and radiation therapy that pronouncedly impair the growth of the tumor through the creation of a favorable antitumor immune microenvironment.

HHP-triggered apoptosis arises from the activation of caspase-3 via extrinsic and intrinsic routes. The extrinsic route is marked through the attachment of Fas ligands to the cell death receptor Fas at the surface of the cell [75]. Cytochrome c tends to be liberated from the mitochondria into the cytoplasm following the activation of the intrinsic signaling route. Apoptosis causes cell death due to cell shrinkage, the disappearance of microvilli, and the condensation of chromatin [71]. The elimination of apoptotic cells is facilitated by “find-me” cues secreted from apoptotic cells to aid the elimination of apoptotic cells via phagocytes [76]. Phagocytes detect the “eat-me” cues on the surface of apoptotic cells and quickly eliminate them. The removal of apoptotic cells stimulates activated phagocytes that release pro-inflammatory molecules, including TGF-β and interleukin-10 (IL-10) [77]. Apoptosis has, nevertheless, been shown to have immunostimulatory properties in some situations, particularly when exposed to γ-radiation or specific chemotherapeutic (CT) drugs, such as anthracyclines [78]. Cell necrosis appears at HHP exceeding 300 MPa [64]. The start of cell necrosis is not contingent on the activation of caspases. Cellular necrosis causes the swelling of cells, the breakdown of organelles, in particular, irreversible injury to mitochondria, and alterations in intracellular ion levels. These alterations eventually result in injury to the cell membranes and the liberation of inflammatory cellular enclosures [71]. Nevertheless, it is not completely understood how far the molecular character of the danger cues of passive exposure of necrotic cancer cells intersects with immunogenic apoptosis.

In their physiological state, apoptotic cells are immunologically inconspicuous or tolerogenic. Apoptotic cells belong to the physiological events that sustain homeostasis inside multicellular organisms [79]. Apoptosis is typified by a number of cellular morphological and biochemical hallmarks, including blistering, chromatin condensation, and the fragmentation of DNA [80]. In opposition to apoptosis, necrosis is linked to inflammation, which is driven through pathological mechanisms [81]. Extracellular high-mobility group box 1 (HMGB1) and heat shock proteins (HSPs) provide characteristic indicators of the immune activator proteins secreted [82]. Moreover, apoptotic and necrotic cells themselves are capable of secreting danger cues [83]. The breakdown of cell membrane integrity results in the liberation of danger signs, which can activate and mature immune cells and frequently causes inflammation. It is important to note that, in the event of apoptosis, the danger signs are altered prior to release, which causes the opposite immunological effect [84]. For example, HMGB1 is commonly oxidized in the course of apoptosis through reactive oxygen species (ROS), and thereby ceases to have an immunological effect [85]. This implies that dying cells and their surrounding tissue define the triggering of immune activation or immunosuppression. HHP can alter the liquidity of membranes and indirectly influence the attachment or conformation of signaling compounds [86]. HHP is also able to alter the forces within the membranes via the augmentation of the bending stiffness to generate biological forces adequate to initiate the mechano-chemical processes [87]. Direct lethal effects of HHP may involve biological membrane injury and other indeterminate, rapid-acting reactions, and ROS generation resulting from biological membrane injury may persist after medical treatment.

### 2.3. Fluid Shear Stress

Shear stress, which is a fluid friction force, constitutes a further key mechanical impulse for the perpetuation of tissue homeostasis. One of the most widely examined cell types in the field of mechanotransduction concerns the vascular endothelial cell, which covers the inside layer of blood vessels. It is known that endothelial cells are capable of perceiving and reacting to alterations in flow direction, pulsatility, and the magnitude of shear forces through mechanosensors and mechanosensitive signal transduction pathways. Consequently, endothelial phenotypes are strongly linked to regional blood flow characteristics and vary in various regions of the vascular branch. Flow patterns have been independently confirmed to regulate the endothelial phenotype by adjusting YAP/TAZ activities: unidirectional laminar flow suppresses YAP/TAZ activities to render endothelial cells silent and inert to inflammatory cells, whereas perturbed oscillatory flow activates YAP/TAZ to enhance a proproliferative and inflammatory endothelial cell phenotype [88,89,90].

Mechanistically, the integrin-Gα13-RhoA axis has been first communicated to convey the flow regulation of YAP/TAZ activities in endothelial cells [88], and disrupted flow via integrin α5β1 has been found to act to promote YAP nuclear translocation and proatherogenic reactions through c-Abl kinase and phosphodiesterase 4D5, respectively [91]. Nevertheless, the mechanisms by which the flow-activated integrin signaling pathways interact with Hippo kinases to modify YAP/TAZ activities have yet to be explored. In complement to integrin-driven mechanisms, short-term unidirectional laminar flow (15 dynes/cm^2^ for 10 min) has been demonstrated to enhance the nuclear localization of YAP in a LATS1/2-independent but angiomotin-regulated fashion [92]. Caveolae, the microdomains of the plasma membrane, are recognized to perceive shear stress cues and have been found to transmit these mechanical signals via the Hippo pathway to regulate the mechanoregulation of YAP/TAZ [93]. Nevertheless, whether the caveolae-dependent mechanism conveys the flow adjustment of endothelial phenotypes is not yet clear. In addition to endothelial cells, multiple other cell types, including mesenchymal stem cells and metastatic cancer cells, are recognized to sense shear stress cues and implement the subsequent biochemical cues in the regulation of cell functions [94,95]. However, there is still a need for additional in vitro and in vivo investigations to prove the function of YAP/TAZ as mechanotransducers in adapting biological performances in the various cancer cell types.

### 2.4. mTOR-FAK Signaling Axis

The mammalian target of rapamycin (mTOR) signaling pathway is involved in the manner in which cancer cells sense physical changes. This causes cancer cells to trigger cellular reactions that promote tumor growth, invasion, metastasis, and chemoresistance. mTOR is at the intersection of multiple signaling frameworks that govern the physical phenotype of cancer cells and transduce extracellular mechanical cues [96,97]. The upstream regulators of mTOR include membrane receptors, integrins, and proteins of the focal adhesion complexes [98], which convey the perception and transduction of mechanical stimuli. Moreover, mTOR-linked cytoplasmic kinases and phosphatases, guanosine tri-phosphatases (GTPases), and transcription factors are also implicated in molecular pathways resulting from divergent physical forces. Upon nutrient accessibility, mTOR stimulates anabolic processes like protein, nucleotide, and lipid biosynthesis, and blocks cellular autophagy (catabolic process in which intracellular macromolecules and defective organelles are recycled by lysosomal breakdown due to various stress factors) and lysosomal biogenesis [99,100,101]. mTOR constitutes two separate complexes, mTORC1 and mTORC2. mTORC1 controls cell growth and proliferation in reaction to growth factors and amino acids, whereas mTORC2 acts in actin organization. In addition, mTORC2 is able to regulate cell proliferation and survival through AKT activation, which is downstream of growth factor signaling. The crosstalk between mTOPC2 and AKT is under the control of NUAK1 [102,103]. In line with this, NUAK1 expression positively correlates with EGFR expression and Akt Ser-473 phosphorylation and malignant progression in several human cancers [104]. The activity of mTORC1 is inhibited during nutrient deficiency, so that the cells can use other sources to obtain nutrients, like autophagy. In addition, mTORC1 is linked to cell death regulation, such as apoptosis, pyroptosis (proinflammatory cues-dependent type of cell-death that is linked to inflammation), and ferroptosis (iron-dependent, non-apoptotic type of cell death) [105]. mTOR is among the signaling routes that have been found to be regulated by integrin trafficking. In ovarian cancer cells, glucose poverty triggers the relocation of α5β1 integrin from peripheral focal adhesions to a more central spot of fibrillar adhesion [106]. This was pinpointed as the major internalization site for fibronectin-bound α5β1, leading to tensin- and Arf4-dependent endocytosis and the lysosomal release of fibronectin-bound α5β1.

Shockwave stimulation has been employed to investigate mTOR-FAK signaling. Following the demonstration of FAK phosphorylation failure after microfilament depolymerization, the upstream regulator has been identified among three kinases reported to be activated in response to pressure-stimulated mechanotransduction, such as GSK-3β, Akt, and mTORC1. Among the three specific inhibitors directed against the latter molecules, solely rapamycin, an inhibitor of mTORC1, blocked FAK phosphorylation, indicating that mTORC1 acts as the upstream controller of shockwave-induced FAK phosphorylation. Moreover, mTOR was determined to be activated by shockwave triggering, with only the mechanotransduction-induced elevation in the number of actin stress fibers, and also the aberrant subcellular localization of mTORC1 as vesicle-like entrapments on microfilaments. These findings indicate not merely a co-ordinated control of FAK phosphorylation by mTORC1 and microfilaments, but also an involvement of mTORC1-FAK signaling in mesenchymal stem cell proliferation [107].

## 3. Mechanical Aspects of the ECM on Cells (Plasticity) and Mechanotransduction

The adherent cells must stick to a solid to be properly shaped and carry out cellular functions. The rigidity of the solid surface (synonymously termed substrate stiffness) can be perceived by cells, such as cancer cells, and challenge cellular behaviors. Inside the human organism, the elastic modulus (the stress–strain ratio that relates to the elasticity of materials) of tissues is capable of differing by more than seven orders of magnitude, from 167 Pa of brain tissue to 5.4 GPa of cortical bone [108,109]. This implies that various cell types favor varying levels of stiffness. Cells can react to changes in substrate stiffness by adapting cell adhesion, spreading, cell phenotype, and migration characteristics. In general, stiff substrates have the potential to considerably facilitate cellular mechanoreaction and mechanosensing due to the elevated intracellular tension that is counterbalanced by the stiff substrate [110]. Fibroblasts grown on stiff substrates, for example, exhibit significantly larger spreads with densely packed actin stress fibers than those grown on soft substrates [111]. On stiffer substrates, the orientations of actin filaments are joint to aligned actin bundles [112].

Cellular plasticity is the phenomenon of cells to adapt broadly to different identities that are linked to a phenotypic spectrum. Despite being a characteristic hallmark of embryonic differentiation, cellular plasticity has also been frequently seen in end-differentiated adult cells exposed to chronic physical and pathological stresses. Cellular plasticity acts as a mechanism of tissue accommodation or regenerative function in such contexts, but it can actually result in the predisposition of tissues toward cancerous transformation.

In multiple adult tissues, cells undergo a switch in identity as being part of a physiological reaction to wounding or an inflammation [113,114]. Such alterations can appear at the scale of individual cells, where the phenomenon is generally termed “transdifferentiation,” or at the scale of a whole tissue, where the transformation is termed “metaplasia.” Metaplasia appears to serve a protective function against chronic damage, either by replacing lost tissue or by forming barriers that better resist adverse conditions. However, in several organs, such as the gastrointestinal tract and other endoderm-derived tissues, the phenomenon of metaplasia is predisposed to cancer. It is crucial to note that metaplasia and transdifferentiation are not equivalent. On the one hand, metaplastic tissues can result from the transformation of one fully differentiated cell type into another, for example, through transdifferentiation. On the other hand, alternative mechanisms, such as selective proliferation, discarding of specific cell types, or changes in the differentiation scheme of stem cells, can also be responsible for metaplastic tissue alterations. Although lineage-tracing efforts in mice have yielded clues to the programs underpinning some types of metaplasia, there is limited knowledge of the cellular and molecular pathways that culminate in metaplasia in humans.

These findings lead to the questions about the very early phases of carcinogenesis, specifically asking why and how tissue metaplasia brings about an escalated carcinogenic risk. Conventional models relying on facile histologic correlations supposed that the ultimate histopathologic phenotype of a tumor would reveal the cell of origin of the tumor. This conclusion may be intuitive; however, the strong relationship between metaplasia and malignancy implies that this simplistic model is erroneous and compels more sophisticated mechanisms to be taken into account (Figure 3).

An appealing hypothesis holds that metaplasia sensitizes cells to the transforming activity of oncogenic cues against which they would ordinarily be resilient. Metaplasia is accompanied with huge alterations in the chromatin scenery, resulting in dynamic modifications in gene expression. These epigenetic and transcriptional alterations allow tissues to deal with acute wounding, but may, at the same time, lay the groundwork for malignant transformation by “opening” more tumor-promoting genes and/or “closing” more tumor-suppressing genes. Structural alterations of this sort in the epigenome can, consequently, confer conducive operating conditions for oncogenes to act in the proper cellular setting.

Mutations in oncogenes and tumor-suppressor genes produce widely varying consequences in diverse cells of origin. For instance, pancreatic acinar cells are susceptible to the transforming actions of mutant KRAS and p53, while pancreatic ductal cells are comparatively resistant [115,116]. Conversely, pancreatic ductal cells are vulnerable to the transforming actions of mutant KRAS and the depletion of PTEN [117]. It is suggested that the probability of tumor development and the ultimate histologic type of tumor, such as acinar vs. ductal carcinoma and cholangiocarcinoma (CC) vs. hepatocellular carcinoma (HCC), is a function of both the precise oncogenic driving agents in place and the cellular compartment wherein they are expressed [118,119,120,121,122]. In keeping with this idea, the lack of the tumor suppressor LKB1 in the germ cells and bronchioalveolar stem cells of the lung not only hastens the KRAS-driven adenocarcinoma of the lung, but also makes the consequent tumors more prone to a switch of lineage to squamous cell carcinoma [123].

Consequently, these studies pose the prospect that the epigenetic and transcriptional restructuring which concomitantly results from metaplasia may itself act as an oncogenic trigger [124,125]. For example, on the one hand, the epigenetic condition of a pancreatic ductal cell at baseline may confer resistance to the oncogenic actions of mutant KRAS and p53 [126], but, on the other hand, its overlay with a pre-existing acinar condition, such as that which would occur during acinar-to-ductal metaplasia (ADM), may convey susceptibility to the same oncogenic cues. In addition, gut microbiota can modify anti-cancer treatment [127]. More research in animal models and human clinical settings—including a thorough examination of chromatin conditions in normal, metaplastic, and premalignant tissues—will contribute to resolving these issues.

As the cancer advances, cancer cells undergo phenotypic and molecular modifications termed cellular plasticity, which can be attributed to microenvironmental cues [128]. Therefore, knowledge of the biophysical characteristics of biomaterials is critical in artificial ECM models and how they control the plasticity of cancer cells (Figure 4). Natural, synthetic, and composite biomaterials that are commonly used as models to encapsulate key biophysical characteristics of the ECM in vitro, comprising architectural and mechanical functionalities, were, hence, first analyzed and catalogued. Special attention has been paid to the impact of various biophysical factors of engineered biomaterials in artificial ECM on cancer cell plasticity, comprising proliferation, morphology, epithelial-to-mesenchymal transition (EMT), migration/invasion, the accumulation of CSCs, pharmaceutical resistance, and the like.

### 3.1. Regulators of Bulk and Local ECM Stiffness and Material Elasticity

A central question emerging from this close interaction between cell and ECM is: how—in an organism with tissue stiffness spanning elastic moduli from ~0.1 kPa (fluids, brain, and lungs) to far into the GPa range (bones)—do cells react to mechanical cues from the extracellular environment to satisfy the requirements of residing tissues while simultaneously regulating the ECM condition to properly self-sustain? The response to this question depends on a delicate equilibrium between “responder” cues from the ECM that control cell behavior (referred to as outside-in signaling) and “effector” cues from the cell that remodulate the mechanical characteristics and/or components of the on-site ECM (referred to as inside-out signaling). Regardless of the direction of signaling, mechanotransduction is fundamental to maintaining a balance between cues from outside and inside the cell. When bidirectional signaling between the cell and ECM is disrupted as a consequence of aging or disease, dysregulated ECM turnover ensues, causing the failure of normal tissue performance [57]. For example, cancer cells evolve complex signaling mechanisms that trigger CAFs to synthesize a rigid tumor stroma that, subsequently, amplifies the events of malignant growth and invasion [129]. At the same time, a cardiac lesion may trigger the hyperactivity of cardiac fibroblasts, resulting in the overdeposition of ECM in the myocardium and, subsequently, cardiac fibrosis [130]. The complexities of mechanotransduction are compounded by ample evidence that mechanical signaling across cell–cell contacts—through the machinery of adherens junctions (AJs)—plays a central role in collective cellular pursuits such as endothelial cell migration and the morphogenesis of epithelia [131].

Nevertheless, it is still very intricate to systematically alter architectural cues, such as pore size and fiber alignment, without modification of the stiffness of the matrix [132,133]. It is known that an elevation in the density causes impaired flexibility of the fibers and, consequently, an elevation in stiffness [134,135]. To overcome this predicament, various strategies have been developed in the field of biomaterials to enable independent manipulation of individual biophysical parameters without altering other characteristics. Macromolecular crowding, which is a perception in which high concentrations of macromolecules capture space and create exclusion phenomena [136], is a potential route to be used to manipulate the fiber architecture while still maintaining the stiffness or density of the matrix. Interpenetrating networks (IPNs) represent combinations of polymer reticulations wherein one network is built in the presence of another [137].

#### 3.1.1. ECM Stiffness Impacts Adhesion, Migration, and Invasion

Cells can perceive physical landmarks over fairly short distances, such as the breadth of a neighboring cell [138]. The cells were able to continually adjust in reaction to the stiffness of the substrate gradient by matching the tissue geometry and applying appropriate tensile forces. This capability is referred to as durotaxis, which plays a role in multiple cellular events [139]. A matrix with stiffness gradients causes cells to move to the stiffer area, which can provide higher tensile forces. In more detail, 3T3 fibroblasts display various polarities and exhibit different orientations at the interface of soft and stiff parts of the matrix environment. These 3T3 fibroblasts can readily migrate over the interface region from the soft toward the stiff area, which, consequently, enlarges the cell-spreading area and increases the exerted traction forces. Cells deposited on the stiff area are not migrating toward the soft area, which implies that they sense the mechanical cues and decide not to propagate toward the soft environment. Instead, they move backwards or even pull back when they arrive at the stiff–soft boundary from the stiff region [140]. Related observations have been made for epithelial cells grown on a microfabricated substrate with a gradient of stiffness. Anisotropic stiffness forces the alignment of actin stress fibers or triggers the formation of focal adhesions of epithelial cells and promotes their growth in the direction of maximum stiffness [138]. Intriguingly, multicellular clusters or aggregates also display durotaxis, and collective durotaxis appears to be vastly more efficacious than that for individual cells. Cells orchestrate their locomotion through active engagement with one another, which permits the fast transfer of force across groupings of cells [139]. The modifications in substrate mechanics affect cell–substrate adhesions, that, in turn, impact cell–cell adhesions. This peculiar mechanical feedback can serve as a tool to elucidate numerous multicellular behaviors, like development, injury healing, and the collective invasion of cancer cells [141].

The reaction of cells to the elements of the ECM relies on mechanics, which superimposes even the action of biochemical cues [142]. In particular, dosage, the kinetics of loading, and the areal distribution of the force govern force transmission and dissemination via adhesion receptors, such as integrins and cadherins, located inside cell membranes [143]. Exposing cells to high forces can ease the mechanotransduction of cells (Figure 5A). It is well-established that a stiff environment encourages intracellular force generation by the spreading of cells and buildup of the cytoskeleton by imposing a high traction force [144]. One perspective states that a loading force exceeding a certain limit can provoke conformational or organizational modifications of the force-bearing proteins, comprising talin, vinculin, FAK, integrins, and stretch-sensitive ion channels [24,26,27,145,146]. These unfolded proteins enlist adhesive and structural proteins to improve force transmission via focal adhesions (Figure 5B). The newly uncovered active sites in these unfolded proteins, in the meantime, enable the enzymes to convert the mechanical signals into biochemical determinants. The composite actomyosin produces adequate traction to equilibrate the intracellular and extracellular force by adhesive proteins. The traction force is transferred alongside the actin filaments toward the nucleus, which governs gene expression and determines cell fate (Figure 5B) [144]. Thus, on stiff substrates (above 30 kPa), mesenchymal stem cells are inclined to develop increased stress fibers and focal adhesions and favor osteogenic differentiation. In contrast, on soft substrates (below 10 kPa), cell adhesion is strongly restrained, and cells go through adipogenic differentiation [147]. The molecular clutch model demonstrates that the propagation of mechanical cues by the clutch is a function of the kinetics of the force loading [148,149]. The stiffness of the substrate normally governs the rate of force loading.

In models of the actin–talin integrin–ligand chain (molecular clutch), the talin unfolding time and integrin–ligand lifetime considerably influence cell adhesion. When a constant force is applied on a talin molecule, the time of unfolding of talin reduces in an exponential manner as the loading force rises (Figure 5C). In the meantime, the integrin–ligand bond lifespan initially grows, and then declines as the loading force gets higher. When lifetimes are sufficient (above a specific threshold for stiffness) for the intracellular force to deploy talin on a stiff substrate, cell adhesion can be converted from a slip-bond to catch-bond regime to established a stable, force-dependent adhesion and activate mechanotransduction routes. The integrin–ligand longevity is strongly prolonged in the catch-bond regime. In contrary, the force-loading rate on the clutch is lazier than the integrin detachment velocity on the soft matrix, which cannot deconvolve talin to constitute a robust adhesion [142,143]. The conformable materials, nevertheless, can change the integrin tethering and detethering kinetics so that the integrin-tethering lifetime is altered, which can lead to the changed mechanosensing of the cells on a soft matrix. Consequently, cell mechanosensation is not as straightforward as the traditional view that a stiff environment could actually provide the stimulation of cellular force and increase the force-dependent response of cells, which is referred to as hard phenotypes.

The ECM in natural tissues is highly dynamic and displays rate-dependent properties like nonlinear viscoelasticity or thermodynamic unsteadiness [150,151,152]. The cells shed and release proteins and enzymes and apply force to rearrange the ECM and adjust the microenvironment to their demands. At the same time, different physical cues of the deformed microenvironment, such as viscoelasticity [153,154], topographic characteristics, and ligand representation, offer multiple incentives to govern cell performance. The conversion of the ECM, thus, represents a force feedback loop for the cells [155]. For instance, the stiffness of the cardiac matrix after a heart attack tends to grow as a result of the development of a fibrotic scar [38]. Skeletal resorption by cell-secreted proteases, such as in microgravity, leads to more permeable and attenuated ECM networks, whereas bone growth occurs by cell-secreted and augmented ECM upon loading [156]. As a result, the mechanical characteristics of the ECM are not invariable, but rather undergo modification over time. This poses new research issues, such as capturing and tracking these dynamic interactions inside such a complex microenvironment, including decoupling the chemical from the mechanical signals and incorporating the dynamic mechanics into the cellular mechanotransduction. Alternatively, ECM stiffness can indirectly alter cell adhesion, migration, and invasion by priming cells for oxidative stress [157].

#### 3.1.2. ECM Stiffness Fosters Drug Resistance

The mechanical ECM cue stiffness can modify the outcome of cancer therapy, as it can interfere in a negative manner with chemotherapy, radiotherapy, and immunotherapy, all of which are altered when the ECM stiffness is challenged due to cancer development and malignant progression.

ECM Stiffness Impacts Chemotherapy

The mechanisms of ECM-based chemoresistance have been revealed. They fall into two main classes: first, physical roadblocks, such as aberrant vascularization and altered matrix stiffness, and second, cell-adhesion-related remedies, such as ECM tissue composition, mechanical signaling pathways, and pro-survival signaling pathways [158]. The impaired restructuring of the ECM leads to enhanced matrix stiffness, vascular collapse, and decreased blood flow, which severely impairs the capacity of medications to penetrate the tumor [159]. Dense fibrosis and abnormal vascularization in PDAC participate in the establishment of a hypoxic and abnormal pH within the tumor microenvironment. Hypoxia adversely impacts drug translocation from the bloodstream into the tumor microenvironment also, and specifically influences the activity of drug transporters and the expression and activity of enzymes that metabolize Phase I drugs [44,160,161]. Glycolysis under hypoxia leads to the formation of large quantities of lactic acid, which lowers the extracellular pH. The capacity of the medication to traverse the hydrophobic membrane is strongly decreased in an acidic environment due to its electrical charge [44]. In parallel with the dense fibrotic ECM that compromises the capacity of medications to propagate from blood vessels to cancer cells, the majority of ECM proteins participate in chemoresistance through the activation of EMT and oncogenic signal transduction pathways, among them, MAPK, PI3K, and YAP [162,163,164,165]. CSCs also represent an influential determinant of chemotherapy resistance, but there are limited research studies focusing on the association between the ECM and pancreatic CSCs (PCSCs) in chemoresistance. Consequently, this part concentrates on the research advancement in ECM-PCSC interactions and chemoresistance. The mechanisms of ECM action in relation to chemotherapy resistance are illustrated in Figure 6.

The physical characteristics of the ECM and the importance of its direct or indirect signaling routes in the survival and maintenance of cellular cells have emerged more prominently [166]. To anchor, the ECM receptor of CSCs also conveys paracrine signals that participate in self-renewal and differentiation events [167]. Hyaluronic acid-CD44 interactions enhance stem cell characteristics, such as NOG and SOX2, and the drug resistance factor (MDR1) expression of PCSCs. Hyaluronic acid synthase 1-3 (HAS1-3) functions as a core enzyme in hyaluronic acid synthesis, and its level of expression is strongly associated with patient outcome. 4-Methylumbelliferone (4-MU) blocks the synthesis of hyaluronic acid and is a registered drug for the treatment of biliary tract disorders [168]. Remarkably, the buildup of hyaluronic acid in mouse models of pancreatic cancer conditioned with 4-MU has been markedly decreased [169]. In addition, other in vivo trials in a mouse model of pancreatic ductal adenocarcinoma (PDAC) have demonstrated that treatment with pegylated human recombinant PH20 hyaluronidase lowers hyaluronic acid levels and augments gemcitabine therapy [170].

The tumor microenvironment (TME) promotes chemoresistance by preserving the phenotype of CSCs. For instance, collagen stimulates the self-renewal of CSCs through integrin signal transduction. JNK signal transduction has been shown to facilitate the upregulation of ECM-related genes. Consequently, the JNK pathway is likely responsible for chemoresistance through the establishment and regulation of the CSC niche [171].

Drug unresponsiveness and chemoresistance are recognized as two of the major drivers of cancer progression, cancer recurrence, and cancer death. Improving insight into the mechanisms whereby cancer cells override chemotherapy-induced cell death and enhance sensitivity to chemotherapeutic agents is essential for enhancing the survival of cancer patients. Nearly all antineoplastic medications are dose-dependent, and the medications penetrate the tumor tissue via the blood vessels, penetrate the tumor stroma, and reach an efficacious concentration of the medication, which is the requirement and pivotal to the elimination of the cancer cells. The expanded volume of the tumor raises the firm tension of the host tissue and causes compression of the tumor’s vascular tissue. The steady elevation of interstitial fluid pressure and the dysfunctional lymphatic vessels induced due to the high permeability of tumor blood vessels increase the pressure exerted on tumor vessels, leading to inadequate blood flow in the tumor, which is not favorable to medication diffusion. Based on this, dense ECM caused by impaired tissue perfusion hinders molecular diffusion, restricts drug permeation, and, finally, decreases the effectiveness of antitumor medications [172].

It has been verified on several animal models that elevated ECM stiffness diminishes chemotherapy-induced cell apoptosis and impairs treatment efficacy, as stromal stiffness compromises chemotherapeutic agent distribution and can cause unresponsiveness to chemotherapy in brain tumors [173] and liver tumors [174]. After doxorubicin application on substrates of 10, 38, and 57 kPa in MDA-MB-231 breast cancer cells, the viability of cells on stiffer substrates turned out to be much stronger, and the blocking of integrin-linked kinase (ILK) abrogated the impact of matrix stiffness on the pharmaceutical effect, indicating that matrix stiffness impacts the chemotherapy sensitivity mechanism process through ILK in breast cancer cells [175]. In the cisplatin-sensitive BRCA2 mouse model of pancreatic ductal adenocarcinoma (PDA), the change in volume of the tumor following treatment with cisplatin demonstrated that the disease stabilized or overtly relapsed, which was associated with reduced tumor stiffness, indicating that the successful treatment response to chemotherapy was associated with diminished tumor stiffness in this animal model [176].

The opposite pattern, though, has been observed in a different breast cancer cell line, MCF-7. However, when equal concentrations of cisplatin and paclitaxel have been applied to a gel matrix of 5.3, 46.7, and 2710 kPa, the viability of MCF-7 cells declined with rising matrix stiffness, indicating that MCF-7 cells on the soft substrate are less sensitive to antitumor medications [177]. In a similar manner, patient-derived human glioblastoma xenograft cells exhibited decreased proliferative activity following temozolomide (TMZ) application with enhanced stiffness in vitro [178]. Similarly, SKOV-3 cells survived more effectively after treatment with 1 µM cisplatin on a 0.5 kPa medium than on a 25 kPa medium. The overexpression of ABCB1 and ABCB4 seems to be associated with the unresponsiveness of SKOV3 cells to cisplatin at low stiffness [179]. In osteosarcoma cells, the cell viability and IC50 value on a 7 kPa substrate were markedly increased following dox-orubicin treatment compared to on a 55 kPa matrix [180]. In HCC cells, it has been observed that they exhibit an enhanced ability to initiate cloning following chemotherapy in a lower stiffness environment, which has been associated with an elevation in positive cancer stem cell markers; among them are CD44, CD133, c-kit, CXCR-4, OCT4, and NANOG [181]. This finding yields a prospective mechanism for the prolonged survival and clone-forming capacity of scattered cancer cells in a soft milieu, such as bone marrow, after chemotherapy.

Vascular permeability of cancer tissue could also be among the mechanisms through which matrix stiffness impacts chemotherapy susceptibility [182]. In cancer tissue, newly sprouting blood vessels are critical to tumor growth and are more convoluted and unripe compared to normal tissue [183]. Heterogeneous blood vessels can result in the inadequate supply of blood to tumor tissue, which, in turn, causes local hypoxia and decreases the effectiveness of chemotherapy and radiotherapy [184]. A stiffer matrix enhances the tension and leakage of arterial vascular endothelial cells, distorts vascular and lymphatic structures in tumor tissue, and compromises vascular functionality, finally resulting in intensified cellular hypoxia, the enhancement of cellular malignancy, and decreased chemotherapeutic agent delivery [185]. The stiffness of the matrix could adjust the activity of MMPs and influence the development of blood vessels in cancer tissues. Decreasing the stiffness of tumor tissue by the administration of the matrix crosslinking enzyme lysyl oxidase (LOX) results in a marked decrease in blood vessel production in a mouse model of spontaneous mammary tumors [186].

Liver tissue stiffness and elasticity due to cirrhosis is a major risk contributor to liver cancer, and sorafenib is the default therapy for patients with advanced hepatocellular carcinoma [187]. Huh7 cells grown on a medium of 4 kPa exhibited resistance to sorafenib compared with a medium of 0.7 kPa. The knockout of YAP potently abolishes sorafenib drug resistance in Huh7 cells cultured on a 4 kPa substrate [188]. In a similar manner, insensitivity to sorafenib on a stiffer substrate has been found in breast cancer cells [189]. In addition, the composition of the ECM, such as fibronectin and type IV collagen, as well as the matrix stiffness, has been revealed to govern the activity of HER2-amplified breast cancer cells. In more detail, the ratio of the phosphorylation of HER2 dropped with growing matrix stiffness (2.5 kPa vs. 40 kPa). Moreover, it has been seen that stiffness is inversely linked to Lapatinib insensitivity [190].

Liver metastases (LM) represent the key cause of death in about 50–75% of colorectal cancer (CRC) patients. Compared with the primary tumor, a marked elevation in stromal stiffness has been seen in fresh and cryopreserved LM tissue in colorectal cancer (1.5 kPa and 0.3 kPa, respectively). Metastasis associated fibroblast (MAF) activation in LM along with the higher expression of COL-1, *α*-SMA, and p-MLC2 added significantly to matrix stiffening via ECM remodeling relative to the primary tumor. In the meantime, a hypertension disease specification has been seen in MAFs from LM, and qPCR analyses on freshly isolated MAFs compared with liver-derived fibroblasts demonstrated a marked elevation in the expression of all major renin-angiotensin system (RAS) components. Anti-RAS medication, such as losartan or captopril, significantly decreases the activity of MAFs and the matrix stiffness of LM within CRC via the impairment of the YAP/TAZ signal transduction pathway, which, then, in its turn, elevates the efficacy of anti-angiogenic treatment, such as Bevacizumab (Bev). In addition, a combined therapy with Bev and anti-RAS agents extended the total survival of CRC patients who proceeded to LM resection as compared to the group with no RAS drugs + Bev (median survival = 55.87/35.83 months), thus shedding more light on the option of matrix stiffness as a potential new target for tumors. Nevertheless, RAS inhibitors do not alter the stiffness of non-metastatic liver tissue, implying that mechano-based therapy may not be of utility unless cancer cells invade the liver [174]. Therefore, the targeting of CAFs and matrix stiffness modification holds considerable promise as a therapeutic tool to enhance the effectiveness of chemotherapy.

ECM stiffness impacts radiotherapy

Radiation therapy used as adjuvant therapy leads to cancer cell death or the deceleration of tumor growth through enhancing the generation of free radicals and reactive oxygen species and the breakdown of the DNA double helix [191]. Even though the reaction of cancer cells to radiotherapy varies according to the cell type, variations in the makeup and characteristics of the tumor stroma may also account for variations in the tumor’s radiosensitivity. A highly aggressive adenocarcinoma breast cancer cell line (MDA-MB-231) and non-transformed epithelial breast cells (MCF10A cells) have been chosen to investigate the impact of irradiation on both metastatic cells and healthy cells with varying matrix stiffness (1.3 kPa, and 13 kPa) using 2 Gy and 10 Gy irradiation doses, which constitute the daily dose of radiotherapy and the one-time maximum dosage for the treatment of metastases, and time points of 1 and 3 days post-irradiation have been selected. The findings revealed that MCF10A cells exhibited a decrease in spreading area and an increase in migration speed and directional persistence with rising matrix stiffness at both times. Conversely, in MDA-MB-231 cultured at 1.3 kPa, the spreading area decreased when irradiated at 2 Gy, which appeared to be similar to MCF10A cells. In addition, the migration speed of MDA-MB-231 cells exhibited a time-dependent decrease and an enhancement of directional persistence at 1.3 kPa with 10 Gy irradiation. MDA-MB-231 cells grown at 13 kPa displayed the opposite response, significantly enlarging their spreading area in a dose-dependent fashion, and the migration speed displayed a significant decline as a potential result of elevated adhesion. Curiously, irradiation caused attenuated and shorter impacts on MCF10A cells compared with metastatic cells, suggesting that healthy cells have a more powerful capacity to sustain themselves, and the migration speed of both cell lines has been significantly diminished on soft substrate, indicating a radioprotective function of physiological ECM that impairs cell motility and invasion [192]. The mechanisms underpinning the impact of irradiation on cell adhesion and cell motility are intimately linked to signal transduction through integrins and FAK. The upregulation of FAK on a stiffer matrix eases on the one hand the formation and breakdown of focal adhesions and encourages cell migration and invasion, but, on the other hand, not the maturation and development of the cytoskeleton [193]. Similarly, SiHa cells, a cell line of squamous cell carcinoma of the cervix, display stiffness-dependent tolerance to radiation through the modified expression of apoptosis proteins. The post-irradiation annexin expression of SiHa cells has been found to be 68.05% ± 9.80%, 47.26% ± 11.65%, and 25.17% ± 14.68% at 0.5, 5, and 25 kPa substrate, respectively [194]. These findings highlight the significant irradiation impacts on cancer cells and the potential for matrix stiffness to be predictive of the radiosensitivity of tumors.

In contrast, some investigations have demonstrated that the stiffness of the matrix has no influence on the sensitivity of cancer cells to radiotherapy. The response of human prostate cancer cell line PC3 to 2 Gy radiation on traditional cell culture substrates (about Gpa) and decellularized spinach leaves (21.8 ± 3.3 kPa) have been examined by evaluating the short-term DNA damage in cancer cells. Even though matrix stiffness governed the proliferation of cancer cells via the YAP/TAZ pathway, DNA damage has been effectively fixed after 6 h of irradiation under various stiffness settings, and no meaningful distinction occurred in the radiosensitivity of PC3 cells on the two plant scaffold substrates following 24 h of X-ray irradiation [195]. Because the mechanism underpinning the impacts of matrix stiffness on radiosensitivity continues to be ambiguous, the controversial outcomes outlined above could be due to varying radiation doses or different cancer cell types.

### 3.2. ECM Mechanical Stress/Loading and Stiffness Alter Cancer Immunity

The established hallmark “immune evasion” represents an obstacle to the efficacy of immunotherapies. The molecular mechanisms and biological consequences underlying immune evasion are widely characterized, but the contribution of tissue mechanical stresses to these processes deserves additional scrutiny. The TME exhibits physical abnormalities, including raised fluid and solid pressures, which act both inside and outside the TME to drive cancer mechanopathology. Remarkably, cancer cells upregulate canonical immune defense mechanisms, comprising EMT and autophagy, upon reaction to these mechanical stresses. Notably, the induction of autophagy and EMT in cancer cells has been seen under mechanical stress conditions—particularly, fluid shear and solid stress. Investigating and profiling the causes and outcomes of mechanical stress in TME could lead to new approaches to counteract immunotherapy resistance. It has been proposed that decreasing or neutralizing fluid shear stress and solid stress could improve the immune evasion of cancer cells and, subsequently, improve immunotherapy outcomes [196].

#### 3.2.1. Force Can Trigger Therapy Resistance

Within solid tumors, the stiffness of the ECM relies primarily on the composition and structural architecture of the ECM, whereas mechanical/physical forces are transmitted in the course of tumor growth. On the tissue level, the compressed and aberrant tumor blood vessels, together with the augmented accumulation of matrix components in the ECM of the tumor, impede the supply of therapeutic drugs to the inner part of the tumor [197,198]. Concomitantly, the increased intratumoral interstitial fluid pressure (IFP), which decreases to normal values at the tumor periphery, results in a flow of fluid from the tumor cavity into the circumjacent tissues, thereby leaching drugs out of the tumor [172,197,199]. While, at the cellular level, the implications of ECM stiffness on treatment resistance are under active scrutiny, research on force-induced drug resistance is still scarce. In the following, mechanical forces can affect the efficacy of cancer therapies, and the molecular mechanisms that are considered to be responsible for this are elaborated.

#### 3.2.2. Increased IFP-Based Shear Stress and Resistance to Therapy

Liquid (shear stress) and solid mechanical stress are associated with carcinogenesis, invasion [200,201,202,203], and the initiation of autophagy [196]. Two sources of shear stress may be encountered by a cancer cell during its lifespan: firstly, shear stresses produced through interstitial fluid flow within the TME [203,204], and, secondly, shear stresses (hemodynamics shear stress) in the circulation occurring throughout intravasation and circulation [200]. Compared to the stress produced through interstitial flow (0.1–1 dyn/cm^2^), cells in the bloodstream are typically exposed to higher shear forces (1–30 dyn/cm^2^) [205]. When cells are exposed to shear stress, the mechanical state of the cells can be deduced, as cancer cells are, in most instances, more compliant (for example, they display elastic deformations) than non-malignant cells subjected to shear stress [196,206]. Autophagy is initiated in certain cancer cells as a survival mechanism triggered by mechanical stress. For instance, lipid rafts, such as cholesterol- and sphingolipid-rich microdomains of the plasma membrane [206], serve as mechanotransducers that stimulate protective autophagy in HeLa cells when subjected to physiological shear forces (20 dynes/cm^2^) [207]. It has also been noted that even lower amounts of shear stress (~1–2 dynes/cm^2^) trigger autophagic flux in cancer cells. Exposure to 1–1.4 dynes/cm^2^ of shear stress stimulates the integrin/cytoskeletal signaling pathways in HCC cells, resulting in cytoskeletal restructuring and, subsequently, autophagosome development [208]. Although shear-stress-induced autophagy has been proven to increase cell migration and invasion, its link to the immune reaction needs to be explored in detail. Therefore, the direct correlation between shear stress and autophagy-mediated immune evasion remains to be studied. A promising approach may be the inhibition of neddylation and its adaptation [203,209].

Inside the TME, interstitial fluid buildup due to vascular hyperpermeability and lymphatic malfunction results in IFP, which interferes with drug delivery. High IFP levels have also been recorded in patients with cervical cancer and are regarded as an unrelated poor prognostic factor for the return of the tumor following radiotherapy [210]. The direct impact of IFP on cancer cell chemosensitivity, nevertheless, is still scarcely explored. Breast cancer cells subjected to hydrostatic pressure exhibited poor responsiveness to doxorubicin, conferred via the upregulation of the ABCC1 drug transporter and, consequently, lowered intracellular doxorubicin levels [211]. High IFP also produces shear stress on cancer cells through fluid flow. In ovarian cancer cell spheroids, shear stress enhanced the expression of EMT markers, ABCC1 drug transporters, and cancer stem cell markers, with decreased sensitivity toward cisplatin and paclitaxel [212]. Moreover, adherent ovarian cancer cells demonstrated a poor response to carboplatin treatment after being subject to shear stress, by activating EGFR-driven MEK and ERK signal transduction pathways to enhance survival [213]. In agreement with these findings, breast cancer cells subjected to shear stress also demonstrated enhanced motility and drug resistance to paclitaxel [214]. Exposure of breast cancer cells incorporated into a 3D collagen matrix to shear stress has been observed to lead to the expression of EMT hallmarks and poor responsiveness to doxorubicin in a number of cell lines [215]. Transcriptomic evaluation of patients diagnosed with triple-negative breast cancer has revealed that the expression of genes involved in chemoresistance is positively related to a shear-stress-induced gene expression pattern [216].

While less is understood about IFP- and shear-stress-induced drug resistance to specific therapies, shear stress has been found to augment the resistance of sarcoma cells to the insulin growth factor-1 receptor inhibition provided by dalotuzumab [217]. IFP also interferes with the trafficking of immunotherapies into the inner part of the tumor and causes the flux of tumor- and stromal-cell-derived immunosuppressive extracellular vesicles in the direction of the tumor margin. These extracellular vesicles can enlist immunosuppressive immune cells that enhance the progression of the cancer [218]. This review article follows the guidelines for the use of “extracellular vesicles” or “exosomes”, which were established in 2018 [219].

#### 3.2.3. Compression-Based Mechanical Stress/Loading and Effect on Resistance to Therapy

This mechanical stress occurs within the constraints of a tumor and ambient host tissue as the density of cellular (for example, CAFs, cancer cells, and immune cells) and ECM (for example, collagen and fibronectin) constituents rises throughout tumor progression [220]. The solid stress is also partly determined through the stiffening of the ECM as a result of the enhanced resistance to the expansion of the tumor. In vivo measurements of solid stress of tumors are still a difficult task. Nevertheless, it has been shown that this stress in human tumors can vary from 0.21 kPa to 19.0 kPa, with higher values in cancers which are more desmoplastic, such as pancreatic cancer [221,222]. Consequently, the massive strain causes the blood and lymph vessels to be compressed, which restricts the transportation of oxygen and nutrients and raises the pressure of the interstitial fluid.

Mechanical forces arise not only intrinsically from modification of the TME structure, but also extrinsically from the responsive interface of cancer and host tissues resulting from tumor growth that leads to expansion in host tissues. These mechanical pressure stimuli have been associated with changes in the rate of proliferation, increased metastatic potency, and survivorship of cancer cells [223,224,225,226,227,228,229]. Nevertheless, their contribution to chemoresistance remains obscure [230]. Pressure stresses caused by the inhibition of the bipolar spindle assembly have been seen to lead to mitotic stalling, which can ultimately compromise proliferation [226]. Since most current chemotherapeutic drugs specifically aim at proliferating cells, this compression-induced mechanism has the potential to restrict therapeutic efficacy. A different kind of investigation, involving mathematical models coupled with experiments, revealed that high compressive stress produced throughout the growth of tumor spheroids in a constrained agarose matrix resulted in reduced cell proliferation and a lower responsiveness to gemcitabine, which is aimed at actively proliferating cancer cells [231]. Mechanical compression in a 3D environment also led to the invasion of ovarian cancer cells and chemoresistance through the increased expression of CDC42, although the actual undergirding mechanotransduction mechanism has not, at present, been clearly identified [232]. Mechanical stress has also been found to elicit resistance to immunotherapies by activating the PI3K/Akt signaling cascade in cancer cells, which attenuates T-cell-induced apoptosis, elevates PD-L1 expression, and encourages the enlistment of immunosuppressive immune cells, such as Tregs [233]. The implications of compression-induced mechanical stress on the effectiveness of targeted treatments still need to be elucidated.

Immune checkpoint pharmacological inhibitors and adoptive T-cell treatment comprise the two principal T-cell-based immunotherapies in tumor therapy, but a high percentage of patients with solid tumors unexplainably lack responsiveness to these treatments [234]. Lately, it has been postulated that the ECM has a kind of physical resistance to T-cell infiltration and multiplication, and that the growing compactness and stiffness of the matrix may cause obstruction to the infiltration process of CD8^+^ T cells, which is among the potential underlying mechanisms of immune evasion and resistance to cancer immunotherapy [172]. T-cell activation and proliferation are critical increments in the immunization of cancer cells, which are hampered by dense ECM stiffness because it impairs the encounter between T cells and antigen-presenting cells [235]. A lower proliferation activity of T cells has been detected on a substrate at 50.6 ± 15.1 kPa versus a substrate at 7.1 ± 0.4 kPa. [236]. Consistent with this, increasing matrix stiffness resulted in the presence of upregulated Treg markers and downregulated markers of cytotoxic T-cell activity. Afterwards, it has been determined that T cells on a matrix with high collagen concentration are weaker in killing autologous melanoma cells. The low activity of T cells in a stiffer matrix could be linked to autocrine TGF-β signal transduction and requires additional investigation [237]. T cells travel in a range of diverse environments, inclusive of collagen matrices, in the amoeboid mode of migration, and it has been proven that the ECM with high density and high matrix stiffness impedes the capacity of T cells to migrate [237]. Reduced matrix stiffness markedly augmented the migration speed and penetration of T cells and boosted the abundance of CD8^+^ T cells in both the stroma and tumor islands by three- to fourfold within PDAC models [238]. Ex vivo culture tissue sections of lung and ovarian cancers also revealed that collagen fibers adversely influenced the migratory nature of T cells into the center of the primary tumor [239,240]. In addition, the higher PD-L1 protein expression on 25 kPa versus 2 kPa substrate has been monitored in HCC827 lung adenocarcinoma cells [241]. In clinical terms, increased collagen levels and a more rigid ECM shortened survival and resulted in a negative response to PD-1 inhibition in melanoma patients, which was associated with a decline in the overall CD8^+^ T cells and an elevation in the depleted CD8^+^ T-cell subpopulations [242]. Tumor-associated macrophages (TAMs) constitute an additional immune factor that has been proven to be influenced by the stiffness of the matrix. A stiffer matrix primarily promotes an M2-like phenotype by fostering polarization events that is recognized as a pro-tumorigenic type of TAM [243]. M2-polarized macrophages comprise anti-inflammatory cells that express hallmarks including IL-10, TGF-β, and ARG1 and are capable of diminishing a strong anti-tumor immune reaction. While differentiating from monocytes to macrophages or while polarizing to an M2-like phenotype, TAMs are generally found in intimate proximity to collagen in the TME and are found to be rather more prone to evolve an anti-inflammatory phenotype when grown on a stiffer matrix [244]. The macrophage and T-cell combined culture revealed that macrophages grown in high-density collagen suppressed the proliferation of T cells to a greater extent than those grown in low-density collagen [237]. The elevated M2 polarization of macrophages has also been monitored in a mouse model with partly higher collagen density and tumor matrix stiffness, and these mice invariably displayed enlarged tumors and more pronounced metastasis [193]. The mechanism underpinning M2-type macrophages and enhancing matrix stiffness appears to be conveyed by the buildup of collagen and its subsequent phagocytosis and consequent lysosomal signal transduction [245].

In summary, the raised matrix stiffness caused through the excessive accumulation and alignment of collagens impairs the performance of immune cells and restricts T-cell-based therapy, all of which results in the immune escape of cancer cells, when the tumor advances to a malignant state and, subsequently, a poor effectiveness of immunotherapy. Based on these findings, an important frontier in the field is to envision strategies aiming at cancer fibrosis to revert the exclusion of immune cells from the primary tumor and to enhance T-cell-based immunotherapy approaches [238].

#### 3.2.4. Targets for Stiffness Regulation in Tumors

Targeting the activation of CAFs is another approach to indirectly target tumors. The CAF-targeting strategy is to impair the CAF-driven restructuring of the ECM that envelops the tumor, and, thereby, the strategy is based on an overall decrease of the stiffness of the tumor. For instance, vismodegib, sonidegib, glasdegib, or saridegib, which target the Hedgehog pathway activation in CAFs through the impairment of a 7-transmembrane protein, smoothened (SMO) receptor [246] (Figure 7), and pirfenidone, which targets the TGF-β pathway activation in CAFs, have both enhanced the effectiveness of chemotherapeutics and immunotherapies in several cancer types, including breast, ovarian, and pancreatic tumor models in vivo [246,247,248]. Vismodegib has received FDA authorization for the treatment of basal cell carcinoma, but additional studies are required for the treatment of all other solid tumors hallmarked by high CAF density, such as breast and pancreatic cancers. Moreover, CAF-targeted treatments comprise the repurposing of various FDA-approved medications, among them are antihypertensives, such as losartan and bosentan, corticosteroid, such as dexamethasone, and antihistamines, for instance, ketotifen and tranilast [249,250,251,252].

Therapeutics interfering directly with the mechanical TME in general are referred to as mechanotherapeutics [253]. In preclinical tumor models of pancreatic and breast cancer, the administration of mechanotherapeutics including pirfenidone, losartan, tranilast, and dexamethasone has been found to decrease tumor stiffness and mechanical forces and enhance tumor perfusion and medication trafficking [248,250,254,255,256]. It is important to note that bosentan is undergoing a Phase 1 clinical investigation in patients with pancreatic cancer (NCT04158635). Losartan has already been successfully evaluated in a Phase 2 clinical trial, which showed that, in combination with radiation and chemotherapy, it converted 60% of unresectable, locally advanced pancreatic ductal adenocarcinomas to a resectable state, turning it into a prospectively curable therapy [257]. Since only a subset of PDAC patients are responders, additional research is required to establish diagnostic biomarkers and to clarify the mechanisms of responsiveness to therapy.

### 3.3. Tumor Growth and Metastasis Is Controlled by Matrix Stiffness

Matrix stiffness, synonymously referred to as ECM stress, constitutes a key physical parameter of the cancer microenvironment and changes during carcinogenesis in response to ECM remodeling through the activation of CAFs and extracellular collagen accumulation, crosslinking, and, consequently, fibrosis. Varying proportions and densities of extracellular collagen within the ECM impart variable levels of stiffness to the matrix. Several solid tumors, including breast and liver cancers, display stiffness of the ECM throughout tumorigenesis, which is frequently accompanied by a characteristic alignment of the collagen fibers [258]. Among those cases, enhancing matrix stiffness revealed a positive link between enhanced tumorigenesis and invasiveness through augmented cell proliferation, cell migration, and invasion [259,260,261]. Accordingly, the stiffness of the ECM matrix is continuously altered by a complex and interactive mechanism that is paralleled by different feedback mechanisms which are advantageous for the development of tumors, but are, at present, not entirely comprehended.

Aggressive breast cancer frequently has an abnormal, stiff peritumoral region due to a desmoplastic response and the infiltration of cancer cells into the peritumoral stroma [262,263]. In comparison to benign breast tumors, the malignant tumor presented a stiffer margin (18.9 ± 18.2 vs. 40.8 ± 43.0 kPa) on the basis of shear wave elastography (SWE) analyses, and the breast cancer with a stiff margin has been enlarged compared with that with no stiff margin, suggesting that the stiffer tumor margin encourages tumor growth and invasion [264]. After cultivating breast cancer cells on stiff (8 kPa) and soft (0.5 kPa) substrates for 7 days and inoculating the cells in mice, the growth of cancer cells on stiff substrate has been more rapid than the continuous growth of cancer cells on 0.5 kPa substrate in the first 7 days. In the stiffer condition group, more activation of the Runt-related transcription factor 2 (RUNX2) gene and concomitant enhanced cytoskeletal dynamics via mechanotransduction through ERK phosphorylation have been observed. The impact of matrix stiffness on cell proliferation can persist and affect cancer cell behavior even when they have metastasized to alternate sites, and the high metastatic capacity arising from high proliferation activity has also been passed down [265].

### 3.4. Regulation of ECM Composition (Ligand Density) by Matrix Stiffness

Modifications of the ECM can be due to the aberrant expression or turnover of matrix constituents, for example, collagen I and III, fibronectin, elastin, tenascins, and hyaluronan, or challenged post-translational modifications, comprising the crosslinking of collagen fibers by LOX [266,267], lysyl hydroxylase, and transglutaminase activity [147], or proteolytic matrix degradation (Figure 8). The latter generates the release of bioactive ECM fragments and ECM-bound factors, all of which may be necessary to overcome cellular constraints, such as barriers for migration and invasion. Moreover, the force-driven physical restructuring of the ECM leads to the reorganization of the ECM organization through the alignment of ECM fibers and, subsequently, opens up hidden passageways for cell migration and invasion. It has been shown that post-translational modifications of ECM compounds alter matrix interactions with other molecules and cell surface receptors, the localization of cells within the tissue, and the breakdown of the ECM [268,269]. Apart from direct ECM remodeling, cancer cells can recruit and activate stromal cells, which are major players in ECM deposition events and are located in the tumor microenvironment via the release of various pro-fibrotic growth factors and inflammatory factors such as epidermal growth factor (EGF), fibroblast growth factor (FGF)-2, platelet-derived growth factor (PDGF) TGF-α, and TGF-β [270]. These stromal cells switch toward CAFs [271,272].

### 3.5. Regulators of Topography

The topography of materials is among the crucial elements that can influence the performance of cells. The cells are incorporated into an ECM that has various topographical characteristics, varying from nanometers to micrometers. Collagen molecules, being the most prevalent fibrous proteins in the ECM, combine to form nano- and microcollagen fibrils and fibers that promote cell adhesion and polarization, and encourage cell migration [8,273,274,275,276]. In addition to the ECM offering numerous topographic landmarks, the cells are periodic and anisotropic in nature themselves. Muscle fibers, for instance, comprise cylindrical, multinucleated cells with tolerances ranging from 5 to 100 μm in diameter [277]. In contrast, cardiac tissue is composed of strongly mustered, rectangular cardiomyocytes that are generally 100 to 150 μm long and 20 to 35 μm wide [278]. In addition, graphene oxide (GO) nanosheets can be employed to decrease cell migration and reduce cell stiffness. Consequently, drug delivery can be increased [279].

In agreement with the texture degree, surface roughness can be grouped into macro/micro-scale, submicro-scale, and nano-scale roughness. Macro- and microtopography commonly afford cell geometric roughness constraints on the order of micrometers to millimeters, and generally cause cells to align with the anisotropy of the ambient microenvironment, which is termed contact guidance of cells [280]. Contact guidance is thought to influence cell polarization and actin cytoskeleton architecture, resulting in the regulation of different cell behavioral traits, including survival, motility, and differentiation. For example, on a convex surface, the cell can deform the nucleus due to the curvature-induced cytoskeletal stress, resulting in the increased expression of lamin A and osteocalcin relative to cells grown on a concave surface [281].

Cells showed slow growth when the surface roughness surpassed the threshold value (Ra = 1 μm). Conversely, increased proliferation has been seen in the cells that have cultivated on surfaces with a roughness of 0.5 to 1 μm [282]. For example, in bone regeneration, for example, macroroughness enhances the frictional connection between implant and bone, ensuring primary implant stability; microroughness provides a broader surface area for bone cells to proliferate and lay down a newly generated bone matrix [283,284]. This behavior may also be important for cancer cells. It is pointed out that the cellular reaction is more complicated at the nanoscale, as the surface characteristics are many orders of magnitude less than those of the cells. On this scale, the size of surface characteristics is comparable to that of individual surface-exposed cell–matrix adhesion receptors, such as integrins. Hence, it could be feasible to address receptor-regulated signaling pathways and regulate cell functionality, involving cell adhesion, differentiation, and self-renewal [285]. The engineering of improved micro/nano-topographic ECM-like biomaterial interfaces is regarded as an outstanding strategy to increase cellular functionality for tissue engineering, regenerative medicine, and cancer treatment.

### 3.6. Regulators of Cancer Cell Transition and Enrichment of CSCs

The concept of EMT is a key biological mechanism underlying the invasion and metastasis of cancer cells. In the EMT course, the cells become devoid of their epithelial characteristics, comprising cell junctions and polarity, and adopt a mesenchymal type of morphology and invasive skills [286,287]. Matrix stiffness potently modulates EMT-related molecular signaling mechanisms to facilitate cancer cell adhesion and invasion. Both MCF10A breast cancer cells and Eph4Ras cells generated polarized ductal acini encircled with an intact basement membrane on a pliable substrate at 0.15 kPa. Conversely, on a substrate at 5.7 kPa, both cells displayed a partial EMT phenotype resembling the malignant phenotype caused by matrix stiffness. The undamaged basement membrane, seen on a substrate with 0.15 kPa, has been disturbed on a substrate with 5.7 kPa. Elevated matrix stiffness fostered the liberation of Twist1 from its cytoplasmic binding partner G3BP2, and the suppression of TWIST1 thwarted the invasive phenotype and stiffness-induced basement membrane unsteadiness at 5.7 kPa, which indicates that the matrix-stiffness-triggered invasion relies on TWIST1. It is even more relevant to emphasize that TWIST1-dependent mechanical transduction, together with TGF-β, has been needed to trigger intact EMT on a stiff medium [288]. Likewise, activin A, which is a constituent of the TGF-β family, has been shown to be abundantly liberated in colorectal cancer, resulting in enhanced matrix stiffness and triggering the ligand-dependent migration of CRC epithelial cells and EMT events [289]. The EMT performance of SiHa cells has been more pronounced on the 20 kPa hydrogel substrate compared to the 1 kPa hydrogel substrate, while the expression of TWIST1 and miR-106b enhanced with an increase in stiffness. The expression of DAB2, which is engaged in the endocytosis of integrin *β*1, has been reduced on a substrate at 20 kPa relative to a substrate at 1 kPa, indicating that matrix stiffness controls the EMT of SiHa cells by targeting DAB2 breakdown through miR-106b [290]. In another investigation, it has been shown that an elevation of matrix stiffness can stimulate the EMT mechanism through TGF-β1 in mammary gland cells from mice and renal epithelial cells from Madin–Darby dogs [291]. Transient receptor potential vanilloid 4 (TRPV4) is a Ca^2+^-preferenced membrane ion channel, and TRPV4 channel activation can facilitate matrix biosynthesis through facilitating calcium influx and, subsequently, the activation of phosphatidylinotol 3-kinase (PI3K) [292,293]. TGF-β1-driven matrix stiffness and EMT events have been significantly inhibited after blocking the TRPV4 channel with a small compound inhibitor [294]. In summary, enhanced matrix stiffness within the tumor microenvironment activates EMT events directly via mechanical transduction pathways and transcription factors such as TGF-β [288]. The entire molecular routes that transmit the mechanical cues from the ECM to the EMT have not yet been identified. A frontier in the physics of cancer research is whether the proper stiffness condition can be an effective regulatory system to sustain stem cells pluripotent, such as cancer stem cells within various types of solid tumor. For mouse embryonic stem cells (ESCs), it has been revealed that ESCs can keep their pluripotency up to 15 passages when grown on a soft substrate (0.6 kPa). Conversely, these stem cells have undergone the loss of self-renewal and pluripotency due to rigid culture environments. It seems to be that stem cells grown on soft gels exert low traction forces on the cell’s architecture matrix (cytoskeleton), which aids the pluripotent preservation of the stem cells [295].

Nevertheless, there are also consequences for CSCs. CSCs represent a small subpopulation of cells with stem-like features in solid tumors that can sustain an undifferentiated stage and the ability to self-renew, and constitute a major contributor to insensitivity and resistance to medications [296,297,298] (Figure 9). The Sox2 gene is primarily concerned with the self-renewal mechanism of CSCs. The expression of the Sox2 gene in Hep-2 laryngeal cells has been detected to be increased on a 1 kPa substrate compared to an 8 kPa substrate, pointing to a stronger stem-like capability of the cancer cells on 1 kPa. In the interim, the ABCG2 protein has been found to be more strongly expressed on a 1 kPa substrate, and has also been identified to be implicated in the generation of the ancillary population phenotype (referred to as stem-like traits), which is strictly related to the unresponsiveness of CSCs toward chemotherapy [299,300]. Osteosarcoma cancer tumor cells cultured on a 7 kPa substrate are less susceptible to doxorubicin (Dox) than those cultured on a 20 kPa and 55 kPa substrate, and exhibit enhanced concentrations of Sox2, Oct4, and Nanog [180]. Moreover, enhanced matrix stiffness has been determined to cause CSC characteristics of HCC Huh7 and Hep3B cells, suggesting increased self-renewal, proliferation, and migration capabilities of cancer cells in a stiff microenvironment. The number of CSC cells also rises as the stiffness of the matrix continues to grow [301]. These findings suggest that alterations in matrix stiffness might have varying implications for the stem-like features of CSC in various kinds of cancer cells, which should be evaluated and debated upon in the future.

### 3.7. ECM Stiffness Regulates Cancer Cell Nuclei Cues

As described before, the nuclear shape is regulated by the stiffness of the substrate on which a cell is grown. Lamin A represents a nuclear sensor for the stiffness of substrates [302]. However, when MSCs are grown on a soft gel, the nuclear envelope shows a highly convoluted nature with scarce lamin A expression. Conversely, cells on a stiff matrix exhibit a plain nucleus and high-level expression of lamin A. The mass spectral assay suggests that lamin A undergoes a higher phosphorylation process in cells cultured on soft matrices, which stimulates lamin A decomposition and turnover [303,304]. Apart from the nucleoplasma and the nucleoskeleton, the DNA-associated molecules fulfill crucial roles in the course of mechanoregulation.

#### 3.7.1. Histone Variants Can Be Altered

The most elementary structure of chromatin is the nucleosome, which is composed of two copies of histones H2A, H2B, H3, and H4 and constitutes the octameric kernel that encases 147 bp of DNA. DNA connecting adjacent nucleosomes is frequently attached to the linker histone H1 [305]. Histone abundance occurs in two ways: firstly, via replication-coupled canonical histones, and, secondly, via replication-independent histone variants. Histone variants act as non-allelic counterparts to the canonical histones and they are also a common characteristic in aged organisms and diseased organisms, as in cancer [306]. These histone variants represent the epigenetic memory. The four major histone variants are H2A, H2B, H3, and H4, which have different tendencies to be diverse and govern specific chromatin areas and gene transcription schemes [307,308].

Histone variants can be inserted into nucleosomes, substituting for canonical histones for a multitude of reasons, among them, the specific chromatin structure. For example, the H3 variant CENP-A is expressed in the centromeres in a tissue- or developmental-stage-specific manner, the H2 variant TH2B is expressed in the testis, or specific genomic functions are required, such as the H2A variant H2A.X, which is required for the response to DNA damage [309,310,311]. Apart from structural and functional aberrations, nucleosomes with histone variants display alterations in nucleosomic mechanics compared to canonical histones. CENP-A nucleosomes appear softer (more malleable) compared to the canonical H3 nucleosomes, as revealed using AFM-based nanomechanical force spectroscopy [312]. In addition, CENP-A nucleosomes can be made stiffer (reduced deformability) by linking kinetochore major constituents, possibly controlling the mechanobiology of the centromere and mitosis. A number of histone variants have also been proven to control the generation of heterochromatin, including H3.3, macroH2A, and H2A.Z [309,310,311]. Whereas H2A.Z.1 has a central function in the development of tumors in the liver, H2A.Z.2 has been described as a driving force in malignant melanomas [313,314]. In total, H2A.Z seems to have a direct involvement in hormone-dependent breast and prostate cancer [315]. The configuration of the histone variants and the attachment partners of the nucleosomes, the fundamental building material of chromatin, could, therefore, have major mechanical effects on the overarching structure of chromatin and the cell nucleus. The selective upregulation of an H1 linker histone variant in plants during leaf development [316] emphasizes the prospective involvement of histone variants in chromatin mechanoreactions. ECM stiffness can remodel chromatin structure, such as altered histone posttranstional modifications. An aberration in chromatin structure can alter the force exertion of cells toward the ECM via the altered linkage to the nuclear lamina [317], which may than impact ECM stiffness.

Histone modifications comprise acetylation, methylation (arginine and lysine), phosphorylation, ubiquitylation, sumoylation, ADP-ribosylation, deimination (converts histone arginine to citrulline and antagonizes methylation of arginine), and proline isomerization, all of which rely on the type of the modification pathway [318]. These changes appear at multiple locations and have diverse regulatory implications, involving DNA repair, DNA replication, transcriptional regulation, alternative splicing, and the condensation of chromosomes. In particular, histone methylation and histone acetylation are key players in epigenetic changes that occur during ageing and cancer development and progression. Transcription becomes activated when H3K4, H3K36, and H3K79 are methylated, while it is suppressed when other histones, such as H3K9, H3K27, and H4K20, are methylated. [319]. In addition, numerous studies have demonstrated that the regulation of organismal longevity and ageing of tissues is linked to the manipulation of histone methyltransferases or demethylases [320]. Hence, several histone changes could be key age-associated landmarks that have the capacity to be used in anti-ageing and anti-cancer pharmaceutical screenings.

#### 3.7.2. ECM Stiffness Impacts the Structure of Chromatin

What is the impact of ECM stiffness on chromatin structure? Mechanical forces can spread rapidly through the cellular architecture over substantial lengths [321]. Contrary to this, diffusible biochemical molecules travel at a slow speed and substantially dissipate with growing distance, even though this can be counteracted through signal boosting via signal propagation cascades [322]. In individual cells, mechanical forces can be perceived and transmitted in two ways: firstly, by a direct force transmission route between the mechanoreceptor and cytoskeleton, and, secondly, by a mechanoreceptor-coupled second messenger biochemical route. The first of these, such as integrins and actomyosin, enables direct force transmission straight toward the cell nucleus [323,324]. The second, such as stretch-sensitive ion channels and G protein-coupled receptors, relies on signaling cascades of a biochemical nature [325] (Figure 10). The fusion of these two routes facilitates rapid and persistent cellular mechanoreactions that frequently modify gene expression [326]. Evolution, of course, possibly favors more than simply velocity, extent, and durability. Mechanoreactions frequently involve memory and plasticity to improve adaptation to future requirements and to assist organisms in adjusting to environmental challenges. [128,327]. Memory and plasticity are remembered and utilized in the shape of epigenetic changes in chromatin and DNA in the cell nucleus, involving alterations in chromatin structure [328,329]. Chromatin is now generally recognized to not only be a passive repository system for DNA, but also have key non-genetic roles, among them, the active adjustment of the mechanical characteristics of the nucleus, such as its viscoelasticity [154,330]. Chromatin structure modification turned out to be a frequent characteristic of the cellular mechanoresponse. Nevertheless, many of its molecular mechanisms and biological mechanisms are not yet clarified. It is also uncertain whether changes in the chromatin structure can mechanically influence the entire cell and possibly also the extracellular surroundings.

Histone modifications, which are post-translational modifications at the N-terminal tails of histones, are a further decisive attribute at the nucleosome level that relates to the higher-order chromatin structure, in other words, the chromatin state. For instance, H3K9me3 commonly has a correlation with constitutive heterochromatin that resides near centromeric and telomeric regions [331,332] and H3K27me3 commonly is associated with facultative heterochromatin that is subject to developmental adjustment [333]. Conversely, H3K4me3 and H3ac, such as H3K9ac/H3K27ac, are commonly found to correlate with euchromatin, which is conducive to gene expression [334]. Increased heterochromatin is linked to nuclear stiffness raise [335]; thus, H3K27me3 seems to be lined to elevated nuclear stiffness. Trichostatin A (TSA) treatment leads to the induction of chromatin decondensation (elevation of euchromatin) and can induce such stem-cell-like characteristics (softer cells). Thus, H9K27ac may be associated with decreased stiffness. Whereas H3K9me3 predates the production of heterochromatin, as the spread of heterochromatin is reliant on the presence of H3K9me3 [336], additional histone modifications provide more complex and interactive insights into the chromatin state. Bivalent chromatin, thus, comprises both H3K4me3 and H3K27me3, which results in an intermediate and equilibrium state existing between heterochromatin and euchromatin [337].

#### 3.7.3. ECM Stiffness Impacts the Linkage of Chromatin with the Nuclear Membrane

The association of chromatin with the nuclear lamina acts as a regulatory determinant for DNA-based nuclear functions. The position of chromosome territories, which represent regions covered by each chromosome within the interphase nucleus, and their connection to the nuclear lamina via the lamina-associated domains (LADs), have a regulatory effect in defining chromatin structure and the transcriptional results of mechanical disturbance [338]. The binding of chromatin to the nuclear lamina is usually accompanied by the suppression of transcription, as the binding of loci to the nuclear circumference causes gene silencing [339] and, subsequently, genetic inheritance [340]. Stress-induced gene expression through integrins is prevented at the nuclear periphery through the silencing of heterochromatin [341]. Subsequently, the heterochromatin state leads to an overall nuclear softening [335,342]. However, a softening of the nucleus by TSA treatment of human MDA-MB-231 breast cancer cells may trigger a stiffening of the cytoskeleton of MDA-MB-231 cells and, consequently, leads to an apparent stiffening of the cell nucleus. The inhibition of actin polymerization using Latrunculin A causes a softer nucleus of MDA-MB-231 cells under TSA treatment [342]. However, the effect is not present in human MCF-7 breast cancer cells, where the TSA treatment caused a softening of the cell nucleus. Thus, it has been hypothesized that the aggressiveness of the cancer cells may account for the differences.

Soft substrate leads to a rearrangement of chromatin from the nuclear core to the nuclear periphery, which is accompanied by the generation of H3K27me3 heterochromatin along the nuclear periphery [343]. Similarly, H3K9me3 heterochromatin rearranges at the nuclear periphery through nuclear deformation brought about by constrained migration. Hi-C data also reveal that a larger proportion of constitutive LADs shifts to the inactive B partition than in other regions of the genome [344,345]. The Hi-C technique is especially important due to the fact that the interaction frequencies it generates can be employed to construct the entire 3-D chromosome and reveal genome structures [346].

Compression by tissue compaction (referred to as crowding) and liquefaction (referred to as fluidization), which is the process of decompressing (referred to as unjamming) cells in tightly compacted tissue, also lead to increased H3K27me3 heterochromatin close to the nuclear periphery [347,348]. In microgravity, chromosome 18, which resides primarily at the nuclear periphery in most human cell lines [349], enhances overall gene expression, whereas chromosome 19, which resides primarily remote from the nuclear periphery, reduces total gene expression, probably as a result of their opposing locomotion (chromosome 18 is traveling inward and chromosome 19 is traveling outward) in the nucleus [350]. In addition, Hi-C data reveal that smaller chromosomes, which tend to be situated outside the nuclear periphery [351,352], exhibit a significant enhancement of intra- and interchromosomal crosstalk under microgravity. Collectively, these findings emphasize the regulatory function of the nuclear lamina connection in force-based alterations of chromatin structure.

#### 3.7.4. ECM Stiffness Impacts Chromatin Structure That Subsequently Changes Nuclear Mechanics and Mechanosensitivity

As sketched in Figure 11, adjacent nucleosomes that are coupled through linker DNA exhibit the traditional beads-on-a-string chromatin structure, which develop into chromatin fibers [353]. Moreover, the chromatin fibers interact with one another to create higher-order structures. At present, the idea of heterogeneous nucleosome arrangements and the folding of the chromatin fiber predominates [354]. Gene regulatory components such as enhancers and promoters are assembled into chromatin loops through local protein interference of the transcriptional apparatus or through cohesin-based loop extrusion within the insulator protein, CCCTC-Binding Factor (CTCF) limits. Chromatin loops then build wider chromatin domains/topologically associating domains (TADs), which may either be dynamic or static. Different TADs of the identical active/inactive state constitute an A/B chromatin partition. Partitions with the identical states create euchromatin/heterochromatin. Individual chromosome territories are generated from multiple partitions within the interphase nucleus [355].

How chromatin loops and TADs control the mechanical characteristics of the nucleus is not yet clear, although recent research has clarified the relationship between chromatin states and the mechanics of the nucleus. Chromatin has been found to impose a mechanical reaction (on a smaller scale) that differs from that of nuclear lamins (on a larger scale) [356]. Heterochromatin stiffens the nucleus, whereas euchromatin softens it [357]. The stretch-driven reduction of heterochromatin leverages its action in softening the nucleus to avoid DNA damage [341]. An elevation of the levels of basic heterochromatin prevents the cell nucleus from stretching and, thus, prevents the softening of the nucleus due to elongation [341]. Differentiated or mechanically dilated bovine mesenchymal stem cells enhance heterochromatin and nuclear stiffness, thereby making these cells more mechanosensitive compared to undifferentiated cells [358]. In addition, changes in chromatin structure have an impact on nuclear mechanics and influence the effectiveness of restricted cell migration [335,342,359]. These findings collectively imply that not only can chromatin structure alter in reaction to mechanical disturbances, but changed chromatin structure can also influence nuclear mechanics and mechanosensitivity, thereby enabling cells to mechanically accommodate to its environment. Intriguingly, a recent work indicates that transcription by itself can function to adjust the nuclear shape even without compromising the nuclear mechanics [360], which further adds another level of how chromatin modifications can influence nuclear structure and integrity.

Equipped with new toolkits engineered to force chromatin loop assembly [361,362,363], or mechanically pull chromatin loops in specific directions and distances [364], future investigations will ultimately be capable of elucidating how chromatin loops and/or TADs influence nuclear mechanics and mechanosensation and how/whether this affects organismal biology temporarily or permanently.

#### 3.7.5. ECM Stiffness Impacts other Nuclear Properties

The remodeling of the chromatin structure relies also on the remodeling of the nucleosomes. As the ECM stiffness impacts the histones and modulates their mechanical properties, such as stiffness, the mechanical cues of the cell nucleus are altered and the cell may be able to squeeze through smaller pores or space in the tissue matrix environment. There is additionally a long-range effect on cellular mechanical cues that is encoded as a mechanocode, such as MESHcode [365], and contributes to the memory of the mechanical stimulation of the entire cells and its organelles. Apart from methylation that directly impacts the transcriptional regulation of genes, the mechanically stimulated cells display a transcriptional signature that relies on the availability of transcription factors, on the linkage of chromatin to the nuclear membrane, and on the amount of euchromatin (transcription) to heterochromatin (no transcription).

Apart from chromatin, non-coding RNAs (ncRNAs) can have an impact on nuclear mechanical cues and, consequently, also on cellular mechanical cues. What types of ncRNAs exists? The ncRNAs can be generally grouped into two classes due to their length. Those with a length of more than 200 nucleotides are referred to as long non-coding RNAs (lncRNAs), while others belong to the small non-coding RNAs (sncRNAs). The latter comprise microRNAs (miRNAs) and various other transcripts including small interfering RNAs (siRNAs) and piwi-interacting RNAs [366]. LncRNAs play a crucial function in the epigenetic regulation of chromatin assembly by recruiting epigenetic determinants, the promoter-specific monitoring of gene expression, the modification of transcript stability, the inactivation of the X chromosome, and imprinting [367]. Moreover, lncRNAs can act as co-ordinators of structures and are involved in the creation of subcellular organelles [368]. Other ncRNAs like miRNAs are involved in other levels of gene expression regulation. The miRNAs fulfill their regulatory purpose by matching with the transcripts of protein-coding genes to knock down their expression at the post-transcriptional stage [369].

## 4. Cells Sense and Respond to Mechanical Cues of the ECM

As the ECM can remodel the cell’s mechanophenotype, cells, in turn, can modify the ECM through mechanical and biochemical cues, such as mechanotransduction and enzyme secretion, respectively.

### 4.1. Cells Can Alter Organization of the ECM in a Biological Manner

The composition and adhesion characteristics of the ECM can be described by the principle of dynamic reciprocity among the cell and its microenvironment. Thereby, the cells experience many mechanical signals. These signals comprise forces that the cells experience from their environment, including neighboring cells, the flow of blood, or the pressure generated in confined spaces. Additionally, the cells utilize their own force-generating mechanism to investigate the mechanical characteristics of the local tissue. Each of these forces can induce a variety of cellular reactions based on shared principles of “mechanotransduction”, in which cells translate mechanical cues into different intracellular biochemical routes. As the list of cellular and tissue-specific events governed by mechanotransduction expands, it is rapidly emerging that forces can elicit specific reactions according to cell type, cellular circumstance, or the way they are perceived through the cell. To obtain both diversity and specificity in reactions, mechanical stimuli must operate in a similar way to biochemical stimuli, in which differences in ligand identity and concentration, detected by a repertoire of receptors, govern a wide array of cellular functionalities. The intricacy of the cellular reaction, thus, results from the wealth of information that is contained in the physical parameters of the mechanical forces, including their magnitude, direction, and dynamics over time, and the capacity of the cells to capture this information. How are force-transmitting molecules capable of recognizing and reacting to these various physical inputs, and how are these molecular reactions incorporated to govern the cellular fate?

Matrix stiffness (synonymously referred to as the elastic modulus or Young’s modulus) of a substance relies heavily on restructuring, crosslinking, and depositing, together with the breakdown of distinct ECM proteins [370]. The accumulation of ECM proteins, which enclose clusters of gel-like hyaluronic acid structures, confers stiffness to the ECM, and the stiffer structures impart resistance to outside pressure stresses on the primary tumors [258]. CAFs, the primary provider of ECM, alter the tumor microenvironment via the expression of LOX, which induces collagen crosslinking during tumor advancement, which is tightly linked to ECM denseness and constitution. The breakdown of protein crosslinking, in return, results in the decomposition of the ECM and reduced stiffness. Collagen constitutes the most prevalent scaffold protein in the ECM and is a key determinant of ECM strength and elasticity in various tissue types. The build-up of collagen and fibronectin causes tensile stress in the circumference of the tumor [371]. The collagen metabolism is disturbed in the course of tumor advancement, which may be reflected in enhanced collagen expression and storage along with increased MMP activity [372]. In this case, TGF-β, a key cytokine implicated in cancer cell adhesion and metastasis, is primarily involved in modulating the activity of fibroblasts and the crosslinking of collagen layers in the ECM [371]. The upregulation of TGF-β is implicated in the evolution of desmoplasia in tumors and has been utilized as a proxy indicator for ECM stiffness [371,373]. Integrins relay mechanical cues from the ECM throughout the interaction between TME and cancer cells by forming adhesion-plaque complexes and control cancer cell performance through cytoskeletal rearrangement [372]. The activation of the integrin-FAK signaling pathway led to the enhanced stiffness of the matrix and, consequently, increased invasion of glioma cells [374]. In a mouse model, upregulated integrins and focal adhesions (FAs) have been linked to elevated matrix stiffness and a stronger invasive potential of mammary epithelial cells [372]. Both FAs and AJs act as core components of cytoskeletal assembly and structural architecture, and among their roles is to co-ordinate multiple biochemical signaling circuits. Moreover, the involvement of AJs in the perception of mechanical cues between cancer cells has also been verified. As principal sensors of geometric and mechanical restraints emanating from adjacent cells, AJs orchestrate actin and membrane dynamics to regulate a variety of morphogenetic events and sustain the integrity of the boundary in reaction to extracellular stresses [375,376]. E-cadherin, a major AJ protein in epithelial cells, has been proposed to facilitate the responsiveness of cells to alterations in matrix stiffness through the activation of multiple actin-binding proteins (ABPs) [377]. The stability of the AJs also has an effect on the activity of the mechanotransduction cues, with the AJs exhibiting a stable status at high tension and a more dynamic status at lower tension [378]. Several mechanosensitive ion channels (MSCs) implicated in carcinogenesis, termed “oncogenic channels”, may also participate in the generation of matrix stiffness via mechanotransduction, in complement to their participation in the cardinal phenotypes of cancer cells, involving migration, limitless proliferation potency, resilience to apoptosis, angiogenesis inducement, and invasion [379,380,381]. Piezo1, which functions as a pressure-sensitive, cation-selective mechanical channel positioned at focal adhesions, has been shown to control ECM and enhance tissue stiffness through the activation of integrin-FAK signal transduction. A stiffer mechanical microenvironment increased Piezo1 expression and encouraged the aggression of gliomas [374,382]. Although it has been established that the elevated matrix stiffness is a direct consequence of the activation of CAFs and the enhanced accumulation and crosslinking of extracellular matrix proteins, especially collagen, it is uncertain whether this activating event is implicated in all tumorigenesis pathways in various cancer types and constitutes an early event in tumorigenesis. In summary, dysregulated CAFs and aberrant collagen accumulation in tumor tissue resulted in the enhanced matrix stiffness of the tumor stroma, which positively correlates with tumorigenesis and tumor growth.

#### 4.1.1. Enzymatic Modification of the Cancer ECM

The ECM constituents, including collagen and fibrin, can be enzymatically broken down. This enzymatic degradation causes the ECM to liberate matrix-tethered biomolecules to guide cell performances. In the interim, the degradation leads to a reorganization of the scaffold that enables the cell to move and invade. The most important enzymes participating in the reorganization of the ECM in the biological framework are metalloproteinases, in particular, the matrix metalloproteinase (MMP) family and a disintegrin and metalloproteinase with thrombospondin motifs (ADAMTS) family [383,384]. In comparison to naturally sourced biomaterials, the synthetic polymers may provide improved batch-to-batch stability and improved quality control [385]. Several biochemical responses have been used in the preparation of degradable biomaterials, covering hydrolysis, such as esters, anhydrides, and thioesters, enzyme-sensitive decomposition, such as MMP-degradable crosslinkers or peptides, and stimulus-sensitive break-down, such as photodegradable systems [386]. Cells need sufficient room to grow and adequate external mechanical support to initiate and control cell functionality, which is particularly relevant in cancer growth. Matrix degradation is, hence, crucial for cell activity, notably in 3D microenvironments [387]. Undegradable or space-limiting rigid 3D hydrogels inhibit cell proliferation, growth, and osteogenic differentiation because the dense crosslinking meshes do not offer sufficient room. Conversely, the customizable, biodegradable hydrogels efficiently improve cell proliferation and functionality in the 3D environment. Thus, MSCs in degradable gels composed of methacrylated hyaluronic acid hydrogel crosslinked with MMP-degradable linkers displayed chondrocyte morphology and expressed a high level of chondrogenic biomarkers. In comparison, MSCs in insensitive hydrogels demonstrated restricted cell spreading with a circular morphology [388]. Similarly, cells could not propagate inside highly crosslinked hydrogels that are compromised by non-degradable reticulations [389].

However, what is the situation of enzymatic degradation in cancer and inflammatory diseases? In contrast to the excessive accumulation of ECM, extensive ECM conversion is triggered by the incorrect expression or activity of matrix-degrading enzymes. MMPs, ADAMs, hyaluronidases, plasminogen, and cathepsins have been seen in cancer and are signs of chronic tissue break-down. A number of MMPs, comprising MMP1, MMP2, and MMP9, have been demonstrated to be implicated in both the enhancement and inhibition of cancer progression in breast and lung cancer via their actions on ECM remodeling and subsequent impacts on intracellular signaling structures [390]. In osteoarthritis, for instance, the abnormal generation of fibronectin, versican, and laminin causes modified integrin-mediated FAK/Src signal transduction and a consecutive elevation of MMP2 and MMP9 expression, resulting in matrix integrity deterioration and enhanced ECM break-down [391]. Similarly, the enhanced cytokine output accompanying rheumatoid arthritis leads to the enhanced expression and aggregation of integrin receptors, closely linked to the activation of their signaling pathways, comprising the ERK, JNK subfamily, FAK/Src, and PI3K pathways. This results in the enhanced synthesis of matrix-degrading enzymes like MMP1 and MMP3 [392]. Specifically, the phosphorylation of the JNK subfamily in synovial fibroblasts has been associated with the elevated expression of collagenases, which is concordant with the chronic break-down of the ECM in rheumatoid arthritis [393]. In complement to integrin-driven signaling programs, the activation of a variety of other ECM receptors can aid the transmission of extracellular cues in healthy and diseased tissues.

#### 4.1.2. Matricellular Proteins

The wording “matricellular protein” has been proposed by Bornstein in 1995. It means that modular, extracellular proteins achieve their functions through tethering to matrix proteins, cell surface receptors, or other molecules, including cytokines and proteases, which, then, interfere with the cell surface. Matricellular proteins are released into the ECM, and even though they can attach to structural ECM constituents like collagen fibrils or basement membranes, they are not assumed to participate in their mechanical functionalities (Figure 12) [394,395]. In opposition to the continuous availability of structural proteins in the ECM, the expression of matricellular proteins is strictly controlled to fine-tune their roles in the preservation and healing of injured tissues [396]. It is remarkable that matricellular proteins are abundantly expressed throughout development, whereas their expression in adult homeostatic tissues decreases to a minimal level. Nevertheless, the expression of a number of matricellular proteins is triggered in the regeneration of tissue damage, inflammation, cancer, and other diseases [397,398]. An example of the multiple implications of matricellular protein are thrombospondins (THBSs). THBSs encompass an evolutionarily conserved family of extracellular, oligomeric, multidomain, calcium-binding glycoproteins that are known to co-operate with other ECM constituents and cell surface receptors [399].

### 4.2. Cells Can Sense Mechanical Cues Passively When the ECM Exerts a Force onto Them

Tissues can be frequently deformed in shear, elongation, or compression, which are facilitated by either static or cyclic mechanical cues, such as stresses. These mechanical cues of the surrounding ECM environment can be sensed by cells through mechanosensory molecules and receptors. Forces acting on cells and exerted by cells on the extracellular environment lead to tensions and deformations that are perceived by a set of specialized molecules termed mechanosensors. These mechanosensors experience a force-dependent conformational modification that modifies the biochemical functionality of the protein. Forces from the cellular environment are usually first perceived at the cell surface, where the force-producing cytoskeleton also applies tension when it contacts various mechanical surroundings. The adhesion complexes, where cells are connected to the surrounding tissue via FAs and to other cells via AJs, have, thus, turned out to be pivotal hubs in the transmission of forces [400,401]. Cells, nevertheless, have a far wider range of mechanosensors, comprising multiple structurally distinct families of force-sensitive ion channels [402] and receptors for biochemical ligands that react directly to force, such as Notch [403] and plexin D1 [404]. In humans, the pathway comprises the cell surface receptors Notch1, Notch2, Notch3, and Notch4, and their ligands delta-like-ligand (Dll), such as DII1, Dll3, and Dll4, as well as jagged 1 (Jag1) and jagged 2 (Jag2). The Notch receptors and Notch ligands each exhibit an extracellular domain, a transmembrane region, and an intracellular domain. Receptor ligand engagement is able to trigger receptor activation by uncovering a concealed extracellular location in the negative regulatory region (NRR) for peptidases. ADAM10 or ADAM17 split this site (S2 cleavage) to generate the extracellular Notch truncation, which is, thereafter, detected from the γ-secretase complex that splits within the transmembrane domain and liberates the intracellular Notch domain (NICD) from the membrane (termed S3 and S4 cleavage) [405,406]. The liberated NICD is then able to engage transcription factor complexes via CSL, which denotes an abbreviation derived from the species names, CBF1/RBPJκ, Su(H), and Lag-1, and either promote or retard transcription [407], resulting in either a similar cell destiny (lateral induction) or an alternate cell destiny (lateral inhibition). Finally, forces at the cell circumference are transferred by the cytoskeleton to other cellular areas like the nucleus [408], which also comprise mechanosensitive compounds and participate in the cellular reaction to external and intrinsic forces.

Mechanosensors operate through a number of common mechanisms whereby the force-induced conformational alterations influence either molecular interactions or the activity of proteins. Forces are able to directly enhance the protein–protein engagement of mechanosensors through enhancing the lifetime of the linkage, such as catch bond, in contrast to the majority of protein–protein interactions where the lifetime diminishes with force, such as slip bond [409]. In addition, forces can act to shape interactions through protein unfolding or demasking, which can either expose cryptic binding sites (CBS) [23,378] or perturb binding motifs, such as the cytoplasmic tail of the β1 or β7 integrin subunits, and FilGAP to filamin A [410]. The type of cryptic site differs for several mechanosensors, and the forces can also uncover proteolytic sites [411,412] or motifs involved in post-translational modifications [413]. Multiple membrane-associated mechanosensors are controlled by force-driven alterations of membrane tension, for example, by driving the gating mechanism of mechanosensitive ion channels [414]. Ultimately, the forces of the cytoskeleton can also work to stabilize specific structural configurations of mechanosensors like integrins [415]. Mechanosensors frequently constitute bulky multimolecular clusters with combined mechanosensors that are regulated by various mechanisms, of which FAs and AJs serve as prototypical models. Mechanical sensors do not work like ordinary on–off switches; instead, their reaction is dependent on certain force characteristics. Forces can impact multiple areas of the cell, but may have varying magnitudes, directions, and temporal patterns, all leading to a distinct reaction and varying biological outputs. The particular mechanisms of force transmission in the single mechanosensors and their organization inside the cell define the capability to distinguish between these various parameters, as will be explained in the following sections.

#### 4.2.1. How Large Are Cellular Forces?

Cellular reactions to mechanical stimuli including flow, ECM stiffness, and tissue stretch are determined as a function of the magnitude of the forces connected to these stimuli. The magnitude of the force perceived by the cells and the sensitivity of the various mechanosensors in this zone dictate how the cells react to the mechanical stimulus. Even though the molecular principles of force sensitivity are not yet fully understood, a number of mechanisms that enable cells to acquire this knowledge have been elucidated. A molecular rationale for sensitivity to force quantities is mechanosensors that have a threshold for activation force, such as the force needed for CBS to be exposed or the force region where catch bonds are generated. Due to the presence of stable intermediary modes of the force-dependent conformations of the mechanosensors, this level of sensitivity can be even more precisely adjusted. Single molecule force spectroscopy of catch bonds has revealed a minimum of three modes, such as weakly, moderately, and strongly bound, at a variety of force strengths across integrin-fibronectin [416], vinculin-F-actin [417], and VWF-GPIb [418]. What is still to be discovered, however, is whether these conditions actually occur in the cells and whether they are linked to varying amounts of biochemical activity. Intermediate states can also occur in mechanosensors that incorporate multiple force-sensitive domains which deploy at varying force levels. CBS in the various rod domains of talin has been shown to unfold at a force of 5 pN for the R3 domain and 10–25 pN for the other domains [419]. Since these rod domains feature various binding partners, this may increase the multiplicity of mechanotransduction routes as a function of the strength of the forces.

Besides the force size-dependent adjustment of individual mechanosensors, the size sensitivity is created by molecular mergers that comprise several mechanosensors with various activation swells. At the same time, it has been demonstrated for ECM stiffness-dependent mechanotransduction through FAs, which involves concomitant integrin-fibronectin catch binding and talin deployment. Since both processes take place solely in a specific force zone, the stiffness becomes an extremely relevant factor [148,420]. The perception of size could be based not solely on the interplay between the mechanosensors within these molecular aggregates, but on their interlocking functioning. For example, tensile forces can not only enhance the association between actin and *β*-catenin/*α*-catenin at cadherin adhesions [421], but also trigger the liberation of *β*-catenin from cadherin to enable its transcriptional role [422], which can possibly be accounted for by varying force thresholds. Size perception can also emerge at the cellular level by activating several types of mechanosensors that are located at a certain distance from one another at various force levels. For example, this is involved in the various mechanisms of protection from nuclear stress depending on the level of stress, with low levels of stress causing the Piezo-induced softening of the nucleus and high levels of stress also leading to the alignment of cells and their actin cytoskeleton in a cadherin-dependent fashion [341].

Different mechanosensors’ specific susceptibilities enable the design of circuits where cellular sensitivity to mechanical stimuli can be manipulated. For example, various integrin subtype and ligands [423], several constituents of the identical mechanosensor family, such as talin-1 and talin-2 [424], or splice variants of the same mechanosensor, such as for Piezo-1 [425], can react to a multitude of force magnitudes. In addition, the mechanical condition of the actual cell, such as the actomyosin contractility and cell stiffness, affects the way cells react to external mechanical stimuli by influencing membrane deformability or exerting a preload on mechanosensors that reduces their surge threshold toward ectopic forces. These mechanisms also account for the diversity of cellular reactions to variations in force magnitudes and the intricacy of the regulation of the dynamic region and the tenderness of the cells.

#### 4.2.2. How Is the Direction of Cellular Forces Regulated?

Since forces represent vector magnitudes that have not only a quantity but also a direction, they inherently deliver directional cues, unlike biochemical signs that need a gradient. Directionality, such as that arising from the direction of blood flow or the direction of tissue stretch, can lead to anisotropic cellular reactions, producing polarized cellular responses. Thus, the directional tension in epithelia leads to the alignment of cell divisions and collective movement following the tension direction due to mechanotransduction across the AJs [426,427]. In a similar way, the majority of cell types align vertically to the direction of uniaxial elongation, thereby focusing on the anisotropic mechanoreaction and the decomposition of FAs [428]. The regulation of AJ dynamics can also rely on the direction of force, as forces spread vertically to the cell–cell contacts and stabilize the AJs, while parallel shear forces demonstrably lead to their deconstruction [429].

In parallel to the polarization of cell response to directional forces, single mechanosensors can induce various reactions according to the orientation of the forces acting on them. Piezo1 perceives both tensile and compressive forces within epithelia, which can trigger cell divisions or extrusion, respectively [430,431]. Most interestingly, Piezo1 is variably sensitive to each of these opposing forces [432], even though the distinct reactions could also be related to distinct cellular Piezo1 populations and/or the action of the inward calcium flow in compressed and stretched cells, respectively [431]. In addition, it has been demonstrated that several mechanotransduction routes are only enabled in a selective manner when forces are administered in a certain direction. Thus, signal transmission by the mechanosensitive TCR/MHC complex in T cells is only accomplished in an efficient manner when the forces act parallel to the attachment interface [433]. In this sense, only unidirectional shear forces on endothelial cells engage integrins and force-sensitive calcium channels to initiate an adequate athero-protective reaction [88,434,435].

The mechanisms used by mechanosensors to translate directional cues into direction-specific cellular reactions are still relatively obscure. The reason for this could be that the arrangement of the mechanosensors in the cell is asymmetric and/or their activation, such as unfolding mechanism of catch bond or CBS unfolding, is optimized when the forces act in a certain geometry. In fact, it has been proposed lately that the stabilization of the link between actin filaments and adhesion complexes relies on the orientation of the forces generated by actomyosin. The catch-bond interplay of vinculin and actin prefers to arise when the forces are directed towards the minus end of actin [417], and a similar directional asymmetry may govern the linkage between *α*-catenin and actin [436]. In addition, the engagement of vinculin with its CBS in talin and that of other force-dependent engagements are more robust when tensile forces are exerted parallel instead of vertical to the bond interface [437]. This geometry constraint of the force-dependent stabilization of actin interference with cell adhesions distorts the structure of actin filaments. In a similar way, the triggering of mechanosensors through external forces can vary according to their inherent geometry and orientation with respect to the force vector. This organization of the mechanosensors is probably subject to anisotropy, so that only some of the molecules are aligned adequately with the direction of force to become activated, while non-aligned mechanosensors may not react or react less strongly. Crucially, anisotropic forces can also be distributed throughout the cell in an isotropic manner via transmission to the cytoskeletal network [438], and thus the anisotropy of the cytoskeleton is expected to underpin the polarized cellular responsiveness to directional inputs.

#### 4.2.3. How Dynamic Are Cellular Forces?

Forces impacting cells can be fleeting and last on the order of seconds, like acute stresses, or hours and days, like morphogenetic movements or a reorganized ECM. In a similar manner, the cellular mechanoreaction toward these signals take place at different timescales [439,440]. In the course of time, in addition to the fluctuating time duration, the forces can oscillate, e.g., as a result of the pulsating stretching of the arterial walls or the “tugging” effects of the interaction between the cell and the ECM [441]. These oscillating forces lead to various cellular consequences in comparison to static forces, such as the specific activation of cell signaling paths and cellular restructuring through cyclic stretching or hydrostatic pressure [438,442,443,444]. In addition, cells can react to different frequencies of force oscillations, which affects, for example, the degree of cellular orientation to axial loads [445].

Oscillation-dependent reactions can be attributed to the fact that the activation of mechanosensors relies on the dynamics of the force over time. For example, cyclic forces can extend the binding duration of catch bonds in comparison to static forces in that they encourage the transition to a heavily bound condition, as experimentally proven for the α5β1-FN catch bond [446]. Mechanosensors can function as bandpass filters, as the transduction sensitivity changes with the signal frequency. The Piezo has been proven to inactivate quickly following its force-dependent aperture. Consequently, the amplitude of Piezo activity can be changed through repeated forces, and this has been demonstrated to be a function of the stimulation frequency [447]. Talin’s unfolding has recently been found to be synchronized with oscillating forces, although it only operates at specific frequencies [22]. While the functional significance and the underlying structural rationale of these mechanisms are not yet clear, these studies indicate that various mechanosensors can process and convey frequency-dependent mechanical signals in a selective manner.

The loading rate, meaning the velocity with which the forces are exerted, is also a decisive factor for the cellular reaction. For example, stretch levels vary between tissues and are high in fast expanding tissues such as the lungs when breathing in air, and low when morphogenetic movements are taking place. The extent of the forces generated by the cells themselves varies according to the viscoelastic characteristics of the ECM, which can result in varying levels of enhancement of the adhesion and spreading of cells [420,448]. The ability of cell–cell adhesions to sustain mechanical stress by stimulating actin reorganization also influences the rate of stress [449]. These variations in loading rate could directly affect the efficiency of the transduction of mechanosensors, since the unfolding of cryptic sites and the kinetics of engagement of the resulting mechanosensor interactions rely on the loading rate [450]. Ultimately, the capture of the temporal dynamics of forces relies on the time scale over which the forces vary, in relation to the time scale of the activation and inactivation of the mechanosensors and the rate of their turnover. A discrepancy between these time scales would cause the cells to lack temporal knowledge of the forces affecting them, which would result in a completely dissimilar reaction.

### 4.3. Cells Can Interact with One Another

Even though various mechanosensors can trigger specific distinct reactions, they commonly act on the identical cellular processes and are able to orchestrate the reaction. For instance, mechanotransduction via integrins, cadherin-based adhesions, and Piezo governs the progression through several phases of the cell cycle [327,431,451,452,453]. Likewise, the Piezo-driven softening of the nuclei and the E-cadherin-dependent realignment of the cells co-ordinately shield the nuclei from mechanical stresses [341]. Several mechanosensors also operate on the exact same signal path, as shown in detail for the control of the Hippo route (for more detail, see review [454]). In an analogous manner, β-catenin-based transcription is mechanistically activated through its phosphorylation at cadherin adhesions [422], and also through the integrin-based suppression of the destroying complex [455]. Due to these connections, mechanical stimuli affecting various mechanosensors cannot merely trigger analogous biological responses, but also empower diverse mechanosensors to interact and guarantee the robustness (or diversification) of the reaction.

#### 4.3.1. Neighboring Cells

Co-ordination is not just the result of interaction at the stage of the downstream mechanotransduction routes, but the mechanosensors themselves also impact the way in which the forces are shared and converted by each other. This has been investigated in detail for FAs and AJs, between which the force partitioning is compensated by their linkage via the actin cytoskeleton (for more details, see review [456]). Thus, the enhanced stiffness of the matrix, which is perceived from the integrins, also leads to increased tensile forces on the AJs [457], and, conversely, the AJs act to alter the tensile forces experienced by the integrins [458,459,460,461]. Piezo has recently been found to be linked to FAs and is enabled at points of traction [462,463]. Inversely, Piezo supports the generation of traction forces through FAs and contributes to their sensitivity with respect to substrate stiffness [374,463]. Numerous additional instances of the interaction of single mechanosensors influencing the regulation and function of others both locally (within one complex) and in a distal manner (spanning several complexes, such as adhesions and nucleus) have been revealed [464,465,466,467], which constitute the complexity of the cellular response to mechanical cues.

In addition to the mutual influence of various mechanotransduction mechanisms, the cellular reaction to mechanical forces also depends on their interaction with biochemical factors, such as growth factors. Since mechanotransduction results in the translation of forces into an intracellular biochemical reaction, the forces will act on analogous pathways and cellular events that are governed via these growth factor cues. In addition, forces are able to modulate the identical receptors that are activated through biochemical ligands that regulate the activity of the receptor either at the engagement level of the receptor ligand itself, such as for EGFR [468], and TGF*β*-R [469], in a ligand-independent fashion, such plexin D1 [404] or possibly both, such as Notch [464,465]. Mechanical and biochemical impulses can synergistically activate downstream signaling processes. In contrary, several receptors exhibit specific downstream pathway signaling in reaction to mechanical activation [464] or initiate different signaling routes when activated in response to mechanical stimuli or their biochemical ligand [404].

The biochemical reaction triggered via the mechanosensors can have the effect of regulating the original mechanical stimulus. This biochemical feedback can occur by weakening the force on single mechanosensor molecules, for instance, by inducing FA growth, or by initiating a cellular reaction that cancels the original forces, such as via enhancing proliferation and, subsequently, decreasing tensile forces. In addition to this level of complexity, biochemical processes can influence the cellular force generation mechanism. This can weaken cellular sensitivity to mechanical stimuli [470], and also spread mechanical forces in the tissue, as demonstrated recently through the reciprocal regulation of ERK activity and tensile forces between adjacent cells [471].

#### 4.3.2. Distant Cells (via Traction-Induced ECM Displacements)

Many cancer cell types exert substantial tensile forces on the enveloping matrix, causing alterations in the ECM that can spread over long distances of tens of cell diameters [52,273,472,473]. When they are embedded in fibrous biological hydrogels like collagen or fibrin, the cells constrict, thereby restructuring and compacting the adjacent ECM fibers. Over a period of a couple of hours, this restructuring can, then, create a visible fibrous band of aligned and compacted fibers that can mechanically connect neighboring cells and impact the internal molecular status of the cells [474] and their active responsiveness [475]. This type of long-range power transmission from cell to cell via the ECM can be regarded as the transfer or sharing of knowledge between cells and, thus, is referred to as cell–ECM–cell communication. This type of long-range mechanical cell–ECM–cell communication has been found to co-determine multiple biological events, for instance, tissue wounding [475], fibrosis [476], vascularization, capillary burgeoning [477,478], the folding of tissues [479], and the invasion and metastasis of cancer [472,480]. In vivo, fiber alignment bands may act as ECM ‘tracks’ for cell movement, which could play a part in wound repair, fibrosis, and cancer metastasis [52,481].

## 5. Emerging New Frontiers in the Field of Mechanotransduction in Cancer

Exploring the expression of mechanosensors like Piezo1 or GPCRs in various tissues is crucial for the design of effective treatments and tactics that enable tumor-specific administration. Genetic analysis of Piezo1 knockout mice revealed a lethal phenotype in utero with disrupted vascular development, emphasizing the need for caution when targeting mechanosensors. The targeted utilization of integrins in cancer therapy has also been extensively examined, but has not yielded any noteworthy outcomes. This may be attributable to the pivotal role of integrins in normal physiology and the absence of a therapeutic opportunity window. With the advent of additional therapeutic tactics that focus on biophysical drivers to overcome drug resistance, there is a demand for experimental frameworks to extensively explore mechanistically engineered drug resistance using model systems and clinically meaningful specimens. These types of platforms might utilize 3D cancer cells or patient-derived organoids encased in a sustaining matrix or subjected to mechanical forces (both compressive and tensile) in mechanical compartments, such as bioreactors. Cancer-on-a-chip models that enable the accurate guidance of mechanical impulses and fluid flow can also be utilized to examine mechanoresistance mechanisms and for the testing of new therapeutic approaches. Computational models utilizing trafficking models and gene regulatory circuits also offer a powerful avenue to explore the interaction between forces, and mechanotransduction and pharmaceutical reactions. Ultimately, machine-learning-based solutions hold the prospect of incorporating huge imaging and multi-omics sets of data in the clinic to clinically classify patients on the basis of mechanistically influenced identifiers. Progress in these areas is expected to result in the identification of novel, highly focused treatments that disrupt mechanistically driven drug resistance and improve the effectiveness of existing treatments for cancer patients.

A frontier is the development of a new microenvironmental platform to culture cancer cells or spheroids, organoids, or tumoroids to explore the effect of the outside mechanical cues on cellular behavior. The roughness, especially, needs to be investigated in, firstly, high-throughput assays to probe a wide range of roughness in a systematic manner. Secondly, the protocol for the generation of substrate roughness needs to be standardized and the underlying mechanics are required. Currently, nano/microarchitectural interfaces have been shown to more efficaciously regulate cellular reaction and bring about the structural and functional organization of cells and tissues [482]. Apart from the culture environment and the dimension of the cell or tissue culture, another frontier is that the organoid culture models often lack vascular systems. Thus, the conditions are not precisely resembling that seen a solid cancer. The development of a vascular network with a prominent lumen is still a major frontier in physics-based cancer research.

Stiffness effect analysis is another frontier, since the majority of the published investigations employ substrates that display stiffness varying from MPa to GPa, which far surpasses the stiffness perceived by the cells in vivo (from several Pa to hundreds of kPa). Thus, it seems to be a requirement and a responsibility to determine the pattern of substrates in conjunction with other physical properties, such as stiffness. Stiffness-controllable hydrogels have been created with a wide-scale surface roughness gradient (Ra = 200 nm–1.2 μm) using soft lithography. MSCs are able to perceive and react to topographical features of the surface in a stiffness-dependent fashion. In particular, the high surface roughness (Ra ≈ 1 μm) improved cellular mechanotransduction on extremely soft substrates (3.8 kPa), similar to that on smooth, stiff substrates. When compared to the soft and smooth surface, the cells largely deformed the soft but rough substrates to remodel the adhesive surroundings. This could be due to the larger number of bonding sites and the reduced stiffness resulting from the very rough characteristics. This investigation indicates that the deformable soft substrate can alter the mechanical characteristics locally via rearranging the density/structure of the force-induced polymer scaffolds, thus improving integrin clustering and enhancing cellular mechanotransduction [35]. A similar effect has been seen in soft fibrillar microenvironments. In particular, fibers with lower stiffness can be more easily deformed due to the force transferred from neighboring fibers. This resulted in an enhanced ligand density at the local adhesion sites, thereby increasing focal adhesion and the formation of cues [483]. Most intriguingly, these curved fiber webs have recently been observed to encourage the generation of cell bridges, as the actomyosin filaments are condensed close to the bent edge of the cells. This allowed the cells to produce a higher intracellular myosin-based force compared to the straight fibers [484].

Another frontier is that the shape and structure of the nucleus are strongly impacted by nano- and microtopography. This effect seems to be pronounced in less differentiated cells, whereas differentiated cells may be more robust. For example, MSCs that are grown on micropillar poly(lactide-co-glycolide) (PLGA) arrays confirmed the hypothesis. The deformation of the nucleus has been triggered on the micropillar medium with a height of 3.2 μm and peaked when the height of the micropillar reaches 4.6 μm or more [484]. Additional investigations suggest that the nuclear deformation of cells in confined spaces is governed through actomyosin-based contractility in conjunction with the LINC complex [485,486]. Counterintuitively, the micropillar arrays can cause the deformation of the nucleus, albeit with limited regions of spread. Nevertheless, they can stimulate increased osteogenesis and attenuated adipogenesis within MSCs [484]. It can be speculated that the functions of cells are not perturbed. However, the fundamental principles are still elusive. The repositioning of chromosomal territories due to the substantial self-deformation of cell nuclei may change the gene expression and, finally, impact the ability of cells to differentiate [487]. Apart from cell culture application, mechanosenitivity and mechanotransduction need to be understood on the level of systems.

### 5.1. Systems-Level Knowledge of Mechanosensitive Pathways

In tandem with advances in the characterization of the extracellular scene, the potential impact of intracellular signal transduction in reaction to the mechanoscape needs to be investigated. The fundamental question of how cells transform biomechanical signals into biochemical sequences, which, in turn, can trigger additional biomechanical reactions, still needs to be understood. From a physical point of view, the extent of mechanical coupling between the various elements of the cytoskeleton is still uncertain. From a biochemical point of view, the signaling pathways that control mechanotransduction need to be clarified even further. Since mechanotransduction often leads to alterations in gene expression, this will involve a full appreciation of the multiple drivers that go in and out of the nucleus in reaction to mechanical cues. Knowledge of alterations in chromatin folding and cellular epigenetics and their interference with cell mechanics will also yield benefits. Gaining insights into the extent to which upstream mechanical cues can be reproduced or influenced will facilitate the engineering of medications that are able to stimulate non-mechanical mechanical cue pathways.

### 5.2. Standardization of Mechanotransduction Models and Approaches

There is a need for a mechanobiological toolset that can contribute to the establishment of standardized models to be employed in the investigation of the cell–ECM interplay. New areas need to break new ground in terms of models, techniques, and tooling, which results in the creation, trialing, and refinement of methodologies. For adequate maturation, however, these methodologies need to be reconciled, standardized, and embraced to mitigate discrepancies in experimental findings. Consensus is required for the choice of standard cells and tissues, the production and characterization of support media, and the mechanical settings. Moreover, mathematical models are used to define the mechanical landscape.

Moreover, new tools need to be devised for fields where existing methods are unsatisfactory. Accurate and non-invasive assessments of the local mechanical characteristics of the cells and matrix in situ are required to completely define the mechanoscape, which can strongly fluctuate in reaction to physical or biological cues [488] and to adequately support efforts to reproduce in vivo environments in vitro. A current initial step in this regard involves a multilab project comparing a wide variety of techniques for gauging the elastic modulus of a discrete cell with standardized techniques. High-throughput 3D biomimetic culture systems coupled with advanced microscopy techniques to track the reactions of cells and tissues to mechanical disturbances will enable a deeper comprehension of the co-evolution of cells and the matrix, as recently documented in a system that enables both high-resolution imaging and controllable imposed strain [489]. Rugged in silico systems that use proteomics datasets and mechanical modeling to forecast force-induced conformational variations of proteins may eliminate expensive and time-consuming deformation assays involving atomic force microscopes, laser tweezers, and Förster resonance energy transfer sensors. In association with the creation of these new instruments, the implementation of international online databases of methodologies, models, and protocols is essential for achieving consistency in this field.

### 5.3. Translation of Mechanobiology into Clinics

Although the field of mechanobiology is fairly new, there has been the approach to translate mechanobiology into clinics. There have been two major strategies for the translation: The first one is focused on the cancer cells themselves and the second one emphasizes the TME.

#### 5.3.1. Trials Aiming at Components of Mechanical Sensing and Mechanotransduction

Promising findings have been made with integrin blockers in Phase 1 and 2 clinical trials [490,491,492]. The survival advantage of chemotherapeutic agents in conjunction with mechanical, path-focusing blockers in comparison to chemotherapy on its own is limited, and the results tend to change for the better only in a few exceptions, such as cilengitide, dasatinib, and vitaxin. The targeted use of mechanosensors offers an alternative way to capitalize on the abnormal activation of mechanotransduction routes and rebuild chemosensitivity. For this purpose, the expression of transient receptor potential (TRP) and Piezo ion channels have been found to be correlated with poor patient survival, which turns these mechanosensors into potential therapeutic candidates [493,494]. In breast cancer cells, TRPM2 has been associated with doxorubicin and tamoxifen tolerance [495], whereas the pharmacologic and genetic inhibition of TRPC5-based autophagy has been demonstrated to increase the cytotoxic impact of adriamycin [496]. In addition, compounds that interfere with TRP channel ion homeostasis by modifying Ca^2+^ and Na^+^ entry can trigger apoptosis and impede cell proliferation and migration [497]. In addition, the metabolite of ginseng saponin (termed 20-GPPD) stimulated the Ca^2+^ inward flow through the activation of the TRPC channel, resulting in apoptosis and the halting of the G1 phase in colon cancer cells [498]. It has also been shown that the blockade of TRPC6 and the reduced Ca^2+^ inward flow in gastric cancer cells causes a standstill of the cell cycle [499]. Clinical trials targeting mechanosensitive ion channels have entered the early testing phase. For instance, the TRPM8 activator D3263 is actually being trialed as a stand-alone agent in a Phase 1 dosage scale-up trial (NCT00839631). TRPV6 is another candidate to block the Ca^2+^-based proliferation and metastatic progression of cancer [500,501]. Recently, a Phase 1 clinical study with the TRPV6 inhibitor (NCT03784677) has been performed to evaluate the tolerated doses and assess the safety of this drug in patients with late-stage solid tumors (for more information, see the review [497]).

The inhibition of Piezo ion channels is another target for mechanosensors that has entered early clinical trials. At present, just a handful of inhibitors of Piezo1 with low specificity are at hand, among them, Gd^3+^, ruthenium red, and GsMTx-4 [493,502]. The G protein-coupled receptor (GPCR) family of mechanosensors may also be considered to serve as possible targets for mechanically mediated chemoresistance. In mice with breast and lung tumors, the pepducin P1pal-7, aimed at the interplay of PAR1 with G-proteins, has been demonstrated to inhibit tumor advancement and metastasis [503]. Recently, doxycycline has been found to be capable of selectively blocking PAR1 in multiple cancer cell lines, comprising breast and melanoma cells [504]. In addition, YM-254890 and FR900359 are versatile and specific blockers of GPCR signaling. In melanoma, YM-254890 in combination with MEK blockers resulted in the killing of cancers in vitro and in vivo [505]. Medications that act on GPCRs are undergoing clinical trials, for instance, the CXCR4 inhibitor TG-0054 [506,507].

#### 5.3.2. Trials Aiming at the Normalization of the Tumor Microenvironment (TME)

Agents that impede MMP-regulated ECM degradation have been evaluated in clinical studies to stop the spread of metastases. In preclinical trials, 4-MU and imatinib have been found to decrease hyaluronan aggregation in pancreatic and prostate cancer in vivo [508,509]. The resulting reduction in hyaluronan-based CD44 activation caused diminished PI3K, Akt, and ERK signal transduction, which compromised migration and invasion. PEGylated human recombinant PH20 hyaluronidase (PEGPH20) represents a further form of therapy that has demonstrated an enhanced responsiveness to gemcitabine and doxorubicin in in vivo pancreatic cancer models [170]. In a clinical Phase 3 trial, the treatment with PEGPH20 in conjunction with nab-paclitaxel plus gemcitabine has not led to an improvement in survival [510]. Preventing collagen reorganization is an additional therapeutic approach aimed at ECM. ECM remodeling comprises specifically the crosslinking of collagen fibers, which is conveyed through the LOX enzyme family [511]. Preclinically, hampering LOX in breast cancer models in mice enhanced the therapeutic outcome of doxorubicin through a decrease in tissue stiffness [512]. In the clinic, of course, the combined application of the LOX2 antibody simtuzumab with gemcitabine has not resulted in more favorable treatment results in patients suffering from pancreatic cancer [513].

## 6. Conclusions

The present understanding of the mechanotransduction of cells in a micro/nanospatial microenvironment is somewhat restricted. Most cell–matrix interferences are hypothesized to dynamically change the regional matrix architecture, viscoelasticity, or spread of ligands on the tiniest scale. Therefore, more developed sensors and techniques are required to decipher the interactions between cells and materials with a better spatial and temporal resolution. Among them are a sensor for molecular tension in the piconewton area and an imaging system with super resolution. These techniques enable force dynamics to be measured directly at the cell–matrix biointerface on a molecular level. Field trials in the future could concentrate on the new biosensing strategies to obtain in situ, real-time, and even long-term monitoring of cells in 2D and 3D dimensions.

In addition, there is still a need to increase the understanding of mechanotransduction signaling pathways. Significant advances have also been made in linking physical stimuli and cell signaling. These trials mainly targeted integrin-based mechanosensitive receptors, myosin-based contractility, and YAP/TAZ nuclear positioning. Mechanosensing and mechanotransduction operations are much more intricate. There are still a lot of open questions that urgently need to be answered. How do cells perceive the mechanical forces exerted by their neighboring cells? How do they react to these mechanical forces? What is the role of bioactive macromolecules, such as lncRNA and miRNA, in the mechanotransduction process? How do various bioactive macromolecules work together in the mechanotransduction process? How can different bioactive macromolecules act together in the mechanotransduction event? How can it be explained why, in different cell types, mechanical forces result in diverse cell functions via specific signaling pathways? How can mechanical forces alter stem cell differentiation? Why is the immune cell movement and activation still demanding?

The response of cancer cells to mechanical signals may rely on the genetic modifications that promote tumorigenesis, involving activating mutations of the Ras, EGFR, or PI3K signal transduction routes. The therapeutic outcome of targeted therapies directed against the abnormal activation of any of these signaling pathways could be affected by an interference with mechanotransduction routes. For this purpose, it would be essential to directly examine both the cancer cells and the TME in patient specimens for genetic heterogeneity and mechanical modifications in comparison to normal tissue prior to therapy. The continuous surveillance of ECM stiffness, vascular performance, and matrix reshaping utilizing an ultrasound-based assessment method [514] or a smart MRI approach [515] may both offer a further understanding of the mechanical factors influencing cancer cells. Incorporating this mechanical profiling with multi-omics profiling would enable scientists to investigate how mechanical aberrations in the TME affect signaling reactions and sensitivities to medications. It is important that the large-scale analysis of signaling pathways in surviving cancer cells and other cells in the TME, including immune cells, will result in rational target identification for the design of therapeutic strategies aimed at mechanoresistance mechanisms and enabling personalized treatments.

## Figures and Tables

**Figure 1 cells-13-00096-f001:**
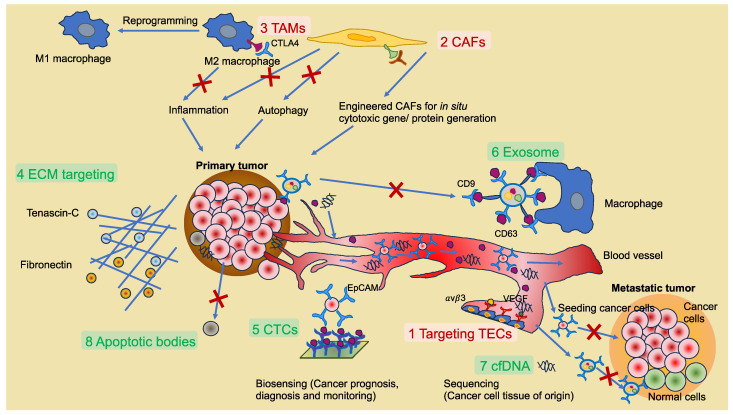
The TME represents various cellular and acellular elements for effective targeting of cancer. Tumor-surrounding cellular targets of therapeutic strategies (1–3) can be tumor endothelial cells (TECs), cancer-associated fibroblasts (CAFs), and tumor-associated macrophages (TAMs) that can be M2 TAMs or reprogrammed M1 TAMs. Besides tumor-surrounding cellular targets, there can be cancer-cell-derived non-cellular targets (6–8), such as exosomes, cell-free DNA (cfDNA) and apoptotic bodies, and circulating tumor cells (CTCs).

**Figure 2 cells-13-00096-f002:**
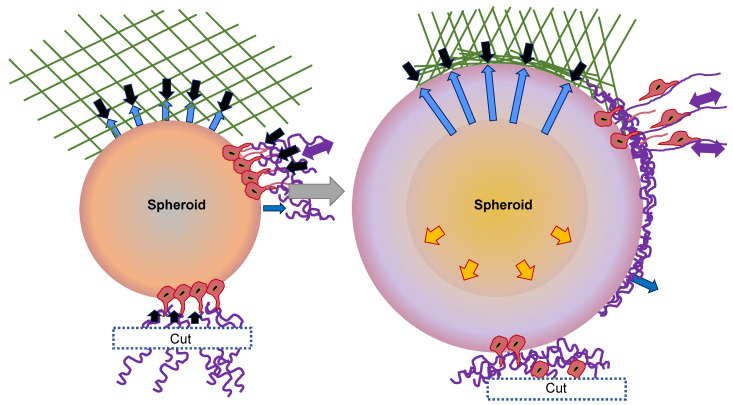
Tension-based invasion mode of cancer cells. The invasion of CT26 colon carcinoma cells out of the spheroid after cutting the collagen gel nearby the spheroid (dashed line). The black arrows show the direction of collagen contraction, as the CT26 cells pull on the ECM fibers (see also text). Left image is shortly after the cut; the grey arrow points to the right image of the spheroid several hours thereafter. The blue arrows illustrate that the expanding spheroid pushes the collagen fibers away. The invading CT26 cells (red) align along the radial fibers and move away from the spheroid. Cells from the underlying layers join in and exert further traction (yellow arrows), which leads to an increase in tension that finally causes the collagen to stiffen and the contraction to subside. When collagen fibers are cut (dotted line) and a free square area is created, the cells in the direction of the cut are unable to generate sufficient tension in the ECM to disengage from the spheroid.

**Figure 3 cells-13-00096-f003:**
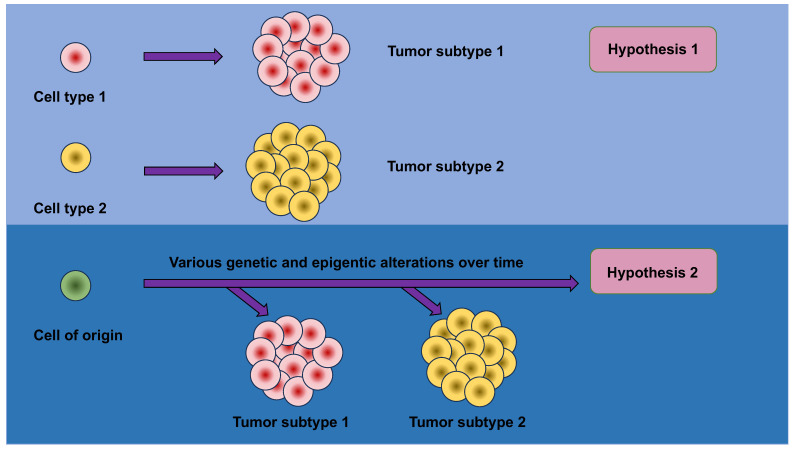
The mechanism of descent plasticity’s contribution to carcinogenesis. Typically, solid tumors are categorized based on the organ in which they originate and their histologic, molecular, and/or transcriptomic profiles. For instance, primary hepatic cancers can be histologically categorized as hepatocellular carcinoma (HCC) or cholangiocarcinoma (CC). Since the cellular origins of the various kinds of tumors continue to be uncertain, there are two universally prevailing hypotheses. The first hypothesis is that the various kinds of cancers originate from differing cells of origin. For liver cancer, this would imply that HCC originates from hepatocytes, but CC originates from cholangiocytes. The second hypothesis states that various kinds of cancers develop in a solitary organ through lineage plasticity, wherein distinct genetic or epigenetic contingencies may predispose a common-origin cell to evolve diverse malignant phenotypes. There exists evidence of lineage plasticity in cancer types, encompassing esophagus (referred to as intestinal metaplasia), liver (referred to as biliary transdifferentiation), lung and cervix (referred to as squamous metaplasia), and pancreas (referred to as acinar-to-ductal metaplasia).

**Figure 4 cells-13-00096-f004:**
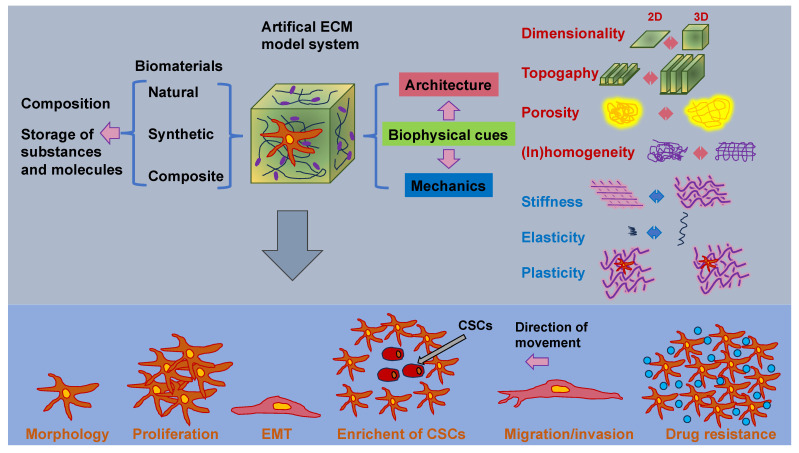
Biophysical properties of biomaterials are increasingly becoming the focus of physical cancer research because of their important role in providing cancer cell function in the context of cancer cell plasticity. EMT = epithelial-to-mesenchymal transition, CSCs = cancer stem cells, ECM = extracellular matrix.

**Figure 5 cells-13-00096-f005:**
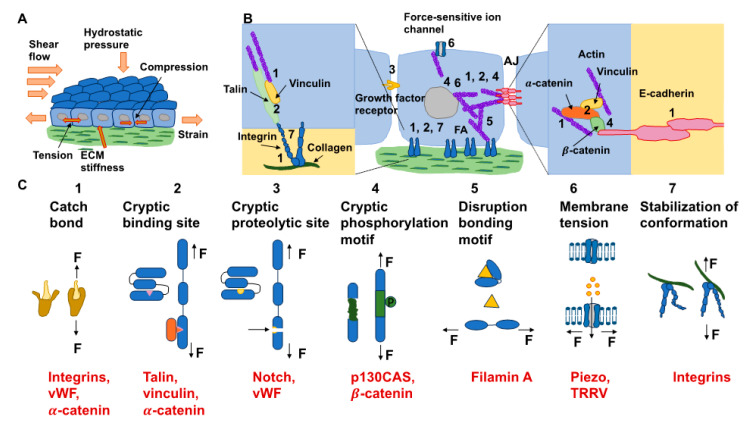
Mechanisms of the mechanotransduction process. (**A**) Cells are subject to diverse mechanical forces and can themselves apply forces to their surrounding environment, which exhibit various mechanical characteristics. (**B**) The kinds of mechanisms of mechanotransduction and possible ones (dotted lines) for mechanosensory proteins and aggregates of proteins within cells are sketched. FAs and AJs that both comprise numerous mechanosensory proteins exhibiting specific mechanotransduction mechanisms (see insets). Even though solely AJs are sketched, it is assumed that other cell–cell adhesions, such as desmosomes and tight junctions, can transfer the forces in a similar way. The envelope of the nucleus comprises distinct proteins and protein assemblies that react to the tension of the nuclear membrane, such as the nuclear pore complex, or they are phosphorylated due to forces, for instance, emerin and lamin. It is still not yet clear whether these proteins are mechanosensors. Even though the gating role of force-sensitive ion channels is driven by force-based alteration in the tension of the membrane, multiple ion channels are directly controlled through forces that are transferred by the linked actin cytoskeleton of the cell. (**C**) Various mechanisms of mechanotransduction over specific mechanosensors, such as catch bonds (1), cryptic binding sites (2), cryptic proteolytic sites (3), cryptic phosphorylation motifs (4), disruption bonding motifs (5), membrane tension (6), and stabilization of conformation (7). The black arrows indicate the direction of the force (F) acting on the molecules or structure. Examples for each of the seven types are shown in red.

**Figure 6 cells-13-00096-f006:**
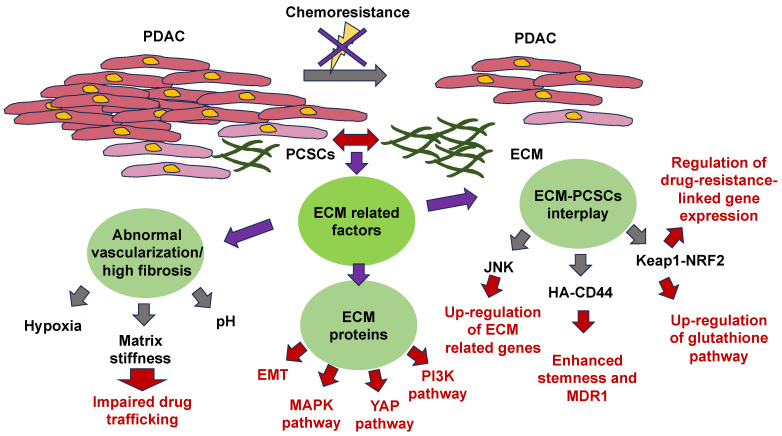
Selected drug (chemotherapy) resistance mechanisms. The ECM-related factors, such as collagen, elastin, hyaluronic acid, tenascin-C, cancer-associated fibroblasts (CAFs), fibronectin, and laminin, can lead to abnormal vascularization/high fibrosis (altered physical barriers) and altered signal transduction due to changed ECM proteins (adhesion-related issues), and altered interplay between the ECM and PCSCs can hinder the effectiveness of chemotherapy. Chemotherapy causes fibrosis and altered vascularization in PDACs that change the pH, and induce hypoxia and stiffness, which affect drug movement. Most ECM proteins promote chemoresistance through activation of EMT and oncogenic signal transduction pathways, like MAPK, PI3K, and YAP. PCSC and their interaction with the ECM are also key drivers of resistance to chemotherapy.

**Figure 7 cells-13-00096-f007:**
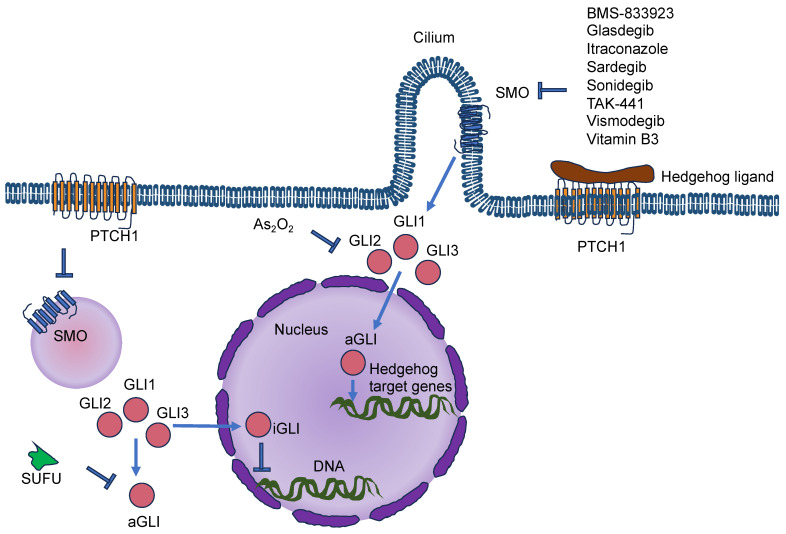
The Hedgehog (Hh) signaling pathway. When Hh ligand binding is not present, the PTCH1 receptor is a negative regulator of the Hh pathway, as it impairs SMO. Upon tethering of Hh to PTCH1, it liberates the SMO. Subsequently, the SMO induces the downstream signal transduction pathway that causes the activation of the transcription factor GLI. As a consequence, the GLI translocates in the nucleus and initiates the Hh-pathway-driven gene expression. Activated GLI (aGLI); inactivated GLI (iGLI).

**Figure 8 cells-13-00096-f008:**
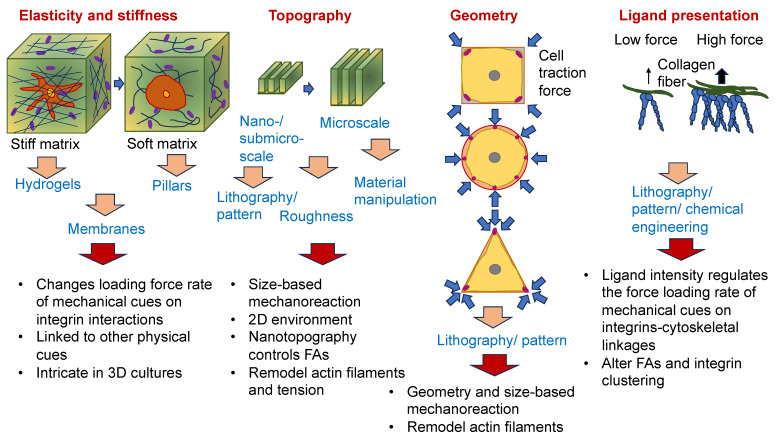
Static ECM cues can be applied to mechanically regulate cancer cells. Models for elasticity and stiffness of microenvironments are hydrogels (soft/stiff) or bending models based on pillars or lipid bilayer membranes (beige arrows). Models for topography can be of different length scales, such as nano-/submicroscale (Lithography/pattern), intermediate scale (roughness of the surface), and microscale (material manipulation) models. The models for geometry denote Lithography/pattern designs. The models for ligand presentation (low and high force) involve Lithography/pattern/chemical engineering. The red arrows indicate what can be modulated and explored.

**Figure 9 cells-13-00096-f009:**
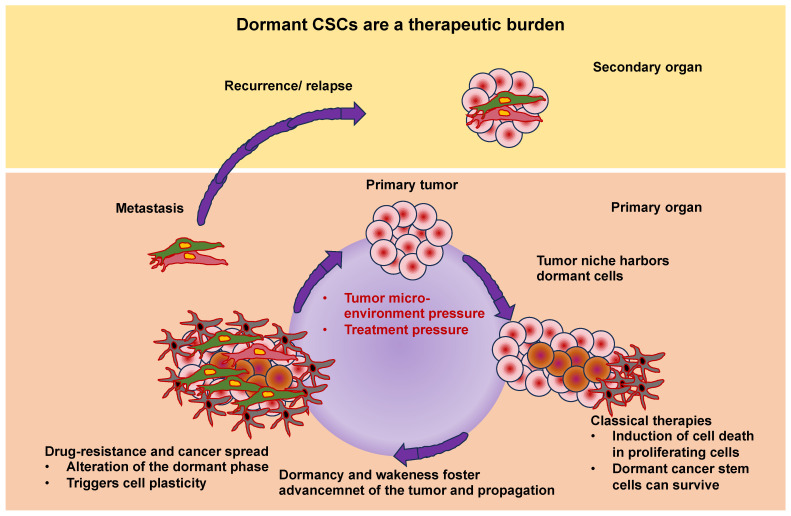
Dormancy of cancer stem cells (CSCs). CSCs possess the unique characteristics to switch into a dormant state, which renders them unavailable for external attack and keeps them as pool of highly proliferative cells, which can regrow and alter the whole tumor, when it is needed.

**Figure 10 cells-13-00096-f010:**
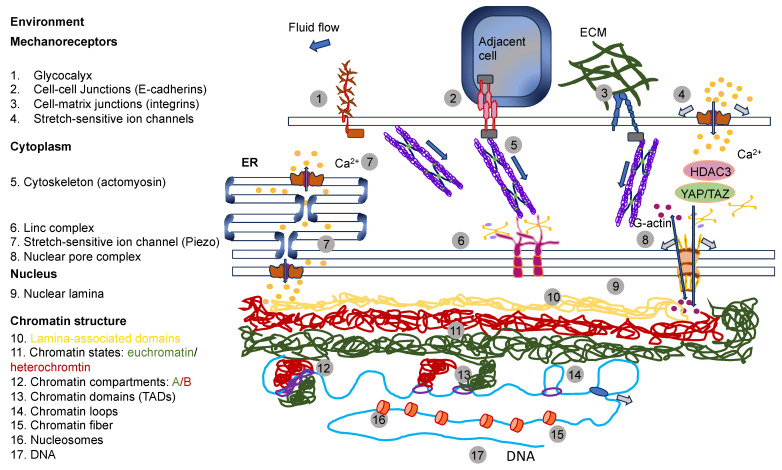
Mechanotransduction routes and the structure of chromatin. Environmental forces are perceived via mechanoreceptors (1–4) on the cell membrane. They transduce forces to the actin cytoskeleton (5) and trigger calcium ion influx (6). The contraction of the actin cytoskeleton transfers forces to LINC complex (7), which is positioned on the nuclear envelope. Thereby, the nuclear envelope deforms and strains the mechanosensitive ion channels, such as Piezzo on the ER nuclear membrane (7). Calcium influx in the cytoplasm and nucleus is induced. Nuclear pore complex becomes dilated (8) that elevates the nuclear import of YAP/TAZ and mechanosensitive transcription factors. Globular (G)-actin and HDAC3 can enter and leave the nucleus. LINC complex (6) transfers forces to nuclear lamina (9), which exerts forces on the chromatin via lamina-associate domains (10, yellow). Chromatin structure comprises chromatin states (11), compartments of chromatin (12), chromatin domains/topologically associated domains (TADs) (13) (lilac ring structures generated by CTCF and cohesion) and chromatin loops generated by CTCF and cohesion (14) or through local transcription mechanisms, chromatin fiber (15), individual nucleosomes (16), and DNA (17).

**Figure 11 cells-13-00096-f011:**
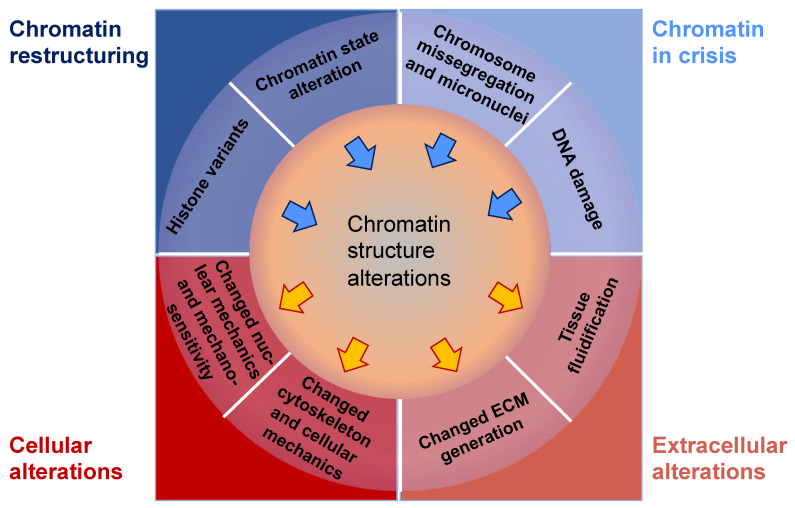
Inside-out mechanobiology through molecular cues impacting the chromatin-structure-based mechanical and structural cues. Principles of molecular cues resulting in changes of the chromatin structure (top, blue) and their mechanical transduction to the ECM (bottom, red). Top left dark blue denotes the restructuring of chromatin and top right light blue stands for the chromatin in crisis. Bottom left dark red illustrates mechanical alterations of cells and bottom right light red denotes the alterations of the microenvironment.

**Figure 12 cells-13-00096-f012:**
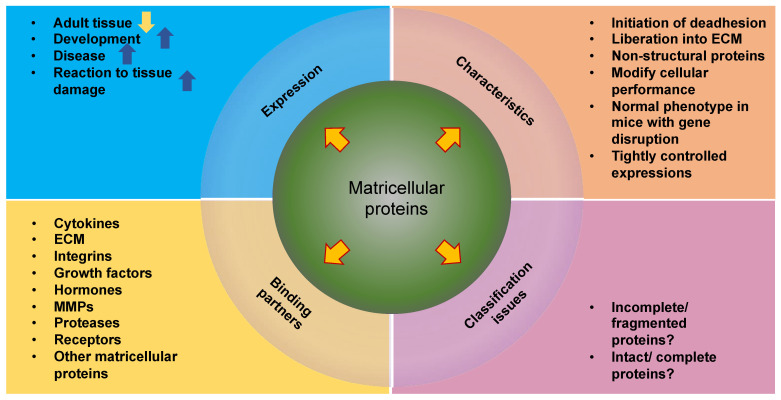
The concept of matricellular proteins. The blue and yellow arrows denote upregulation and downregulation, respectively.

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
