# Peer review of "Extracellular Matrix Cues Regulate Mechanosensing and Mechanotransduction of Cancer Cells"

_cells, 2024, doi:10.3390/cells13010096_

Round 1
Reviewer 1 Report
Comments and Suggestions for Authors
This is a very interesting review, however, there are a few points that have to be fixed prior to acceptance.
1) The abstract should be revised. The current abstract has some facts on interactions between ECM and cells. The latest findings (preferably numerical results) from previous research must be added.
2) The aims and prospects of the review should clearly be described at the end of the introduction.
3) In this kind of comprehensive review, an appropriate graphical abstract is strongly advised.
4) In the 2.6.2 section, the authors claim ECM stiffness impacts the structure of chromatin. My question is: Does ECM stiffness affect chromatin structure or do aberrations in chromatin structure (due to wrong histone PTM) affect ECM stiffness?
5) How does heterochromatin to euchromatin due to H3K27me3 and H3K27ac affect/change the cell’s stiffness? (line 928-940)
6) In the mechanotransduction mechanisms section ( line 1454), the mTOR signaling pathway is very important, but it has not been mentioned in this review at all.
7) It would be nice if the authors opened a section and discussed the mechanosensing of ECM effects during embryogenesis in terms of the second Newton law. On the other hand, how ( based on cell-signaling pathway and physical law) does ECM stiffness affect embryogenesis?
8) I suggest preparing a list of sections and sub-sections at the beginning of the manuscript.
9) This research article may be used to enrich your manuscript about cell migration. https://www.nature.com/articles/s41598-021-95624-0
Author Response
Dear Reviewer 1
I have responded to all your points and included them (see yellow marks).
This is a very interesting review, however, there are a few points that have to be fixed prior to acceptance.
1) The abstract should be revised. The current abstract has some facts on interactions between ECM and cells. The latest findings (preferably numerical results) from previous research must be added.
Answer: I have included them and adapted the abstract.
2) The aims and prospects of the review should clearly be described at the end of the introduction.
Answer: I have rewritten that part.
3) In this kind of comprehensive review, an appropriate graphical abstract is strongly advised.
Answer: It is now included.
4) In the 2.6.2 section, the authors claim ECM stiffness impacts the structure of chromatin. My question is: Does ECM stiffness affect chromatin structure or do aberrations in chromatin structure (due to wrong histone PTM) affect ECM stiffness?
Answer: ECM stiffness can remodel chromatin structure. Aberration in chromatin structure can alter force exertion of cells toward the ECM, which may than impact ECM stiffness. It is now included in the manuscript.
5) How does heterochromatin to euchromatin due to H3K27me3 and H3K27ac affect/change the cell’s stiffness? (line 928-940)
Answer: Heterochromatin is increased and leads to increased nuclear stiffness (Hsia et al 2023), thus H3K27me3 seems to be lined to elevated nuclear stiffness.). Trichostatin A (TSA) treatment leads to induction of chromatin decondensation (elevation of euchromatin) and can induce such stem cell-like characteristics (softer cells). Thus, H9K27ac may be associated with decreased stiffness. It is now also included in the manuscript.
6) In the mechanotransduction mechanisms section (line 1454), the mTOR signaling pathway is very important, but it has not been mentioned in this review at all.
Answer: I agree that it is really important. Thus, I included it in the section of mechanotransduction.
7) It would be nice if the authors opened a section and discussed the mechanosensing of ECM effects during embryogenesis in terms of the second Newton law. On the other hand, how (based on cell-signaling pathway and physical law) does ECM stiffness affect embryogenesis?
Answer: This is a very interesting suggestion, but it should be dealt with in another article, as it is beyond the scope of this article in terms of content and length. The very large size of the article has already been criticized by other reviewers. The stiffness of the extracellular environment plays a role in directing cell division, maintaining tissue boundaries, channeling cell migration and controlling differentiation. Thus, it may function in early embryogenesis after the embryo has divided into two hemispheres such as animal and vegetal poles. However, the individual cell stiffness changes during cell division, where it is increase.
8) I suggest preparing a list of sections and sub-sections at the beginning of the manuscript.
Answer: I have prepared it.
9) This research article may be used to enrich your manuscript about cell migration. https://www.nature.com/articles/s41598-021-95624-0
Answer: Thank you for the citation. It is now included.
Best regards
Claudia Tanja Mierke

Reviewer 2 Report
Comments and Suggestions for Authors
The author outlined the impact of extracellular biophysical properties on cellular processes such as growth, movement, and differentiation. Cells not only respond to mechanical signals but also manipulate matrix mechanics, influencing signal transduction. This provides crucial insights for treating diseases like cancer, revealing the role of mechanotransduction in tumor therapy. However, a significant issue in the article is its excessive length. As a review article, maintaining conciseness is crucial for ensuring depth and quality of content. Lengthy articles may hinder reader comprehension and risk burying important information. Imposing a page limit encourages the author to focus, distill key information, enhancing the article's insight and readability. Another concern is the temporal concentration of references, mainly from five years ago. Incorporating more recent studies would better reflect the latest developments in the field, providing a more comprehensive understanding of the current research status. Additionally, the length of the reference list is also excessive and would benefit from some refinement.
Author Response
Dear Reviewer 2
I have responded to all your points and included them into the manuscript (see yellow marks).
The author outlined the impact of extracellular biophysical properties on cellular processes such as growth, movement, and differentiation. Cells not only respond to mechanical signals but also manipulate matrix mechanics, influencing signal transduction. This provides crucial insights for treating diseases like cancer, revealing the role of mechanotransduction in tumor therapy.
1.) However, a significant issue in the article is its excessive length. As a review article, maintaining conciseness is crucial for ensuring depth and quality of content. Lengthy articles may hinder reader comprehension and risk burying important information. Imposing a page limit encourages the author to focus, distill key information, enhancing the article's insight and readability.
Answer: It is difficult to meet the different requirements of several reviewers, another reviewer even suggested adding more sections.
In my review article, I tried to cover the topic comprehensively in order to arouse the interest of many different readers, which made the article quite long. I have taken on board the reviewers' suggestions to organize and restructure the article more clearly and put a table of contents at the beginning. This makes it easier for readers with a wide range of interests to find the aspects of my work that are important to them and, if necessary, skip the points that are not so important to them or read them only briefly.
I hope this compromise solution meets with your approval.
2.) Another concern is the temporal concentration of references, mainly from five years ago. Incorporating more recent studies would better reflect the latest developments in the field, providing a more comprehensive understanding of the current research status.
Answer: I totally agree. I have now included more recent references.
3.) Additionally, the length of the reference list is also excessive and would benefit from some refinement.
Answer: I have added new references and also followed the suggestions of the other reviewers (reviewers 3 and 4) to add more references (preferentially newer ones). It is now difficult for me to refrain from specific references, as these are also of great importance, even though they were written some time ago. They have already established themselves.
Best regards
Claudia Tanja Mierke

Reviewer 3 Report
Comments and Suggestions for Authors
This comprehensive review article by Mierke is a piece of art. I really enjoyed reading the manuscript and I think it will be of great interest to cancer researchers. The only suggestion I have would be changing the structure; I would recommend that Section 4, which give a general information about mechanotransduction mechanism, is moved to an earlier spot. Additionally, the manuscript is centered around stiffness; mechanical stress/loading is less mentioned throughout the manuscript, which can be improved further.
Author Response
Dear Reviewer 3
I have responded to all your points and included them into the manuscript (see yellow marks).
This comprehensive review article by Mierke is a piece of art. I really enjoyed reading the manuscript and I think it will be of great interest to cancer researchers.
1.) The only suggestion I have would be changing the structure; I would recommend that Section 4, which give a general information about mechanotransduction mechanism, is moved to an earlier spot.
Answer: I agree and moved in to an earlier spot.
2.) Additionally, the manuscript is centered around stiffness; mechanical stress/loading is less mentioned throughout the manuscript, which can be improved further.
Answer: It is a very good point and is now included in the manuscript.
Best regards
Claudia Tanja Mierke

Reviewer 4 Report
Comments and Suggestions for Authors
The author provides a long and thorough review of the role of the extracellular matrix in cancer development and survival. They also explain in detail how mechanosensing and matrix stiffness are related to chemotherapy resistance, tumor progression, and differentiation. The references are updated and follow the state of the art from the field. Some suggestions to improve text quality are described below:
1. Double-spacing in line 62.
2. Text is found duplicated in line 134 - "... on the one hand, but on the other hand, ...".
3. The paragraph ranging from lines 182 to 190 requires at least one citation.
4. Lines 208, 210, 225, and 226: please substitute the hyphens (-) with Em Dashes ( – ).
5. The design of Figure 3B is too small and crowded. I would suggest increasing the figure size to better display both the text and the diagram.
6. Figure legends for Figures 3C, 4, 6, and 12 should be expanded to include a more detailed description.
7. Line 563: Following the MISEV2018 guidelines (Théry et al, 2019 - PMID: 30637094), researchers should refrain from using the term "exosomes" unless the endocytic pathway of those extracellular vesicles is proven. If the original paper does not verify this biogenesis, authors should avoid this term and apply the broader "extracellular vesicles (EV)" terminology.
8. Lines 691-692, 851-852, 873-874, 888: check paragraph formatting.
Comments on the Quality of English LanguageAlthough the science is sound, I suggest an extensive review of the English language to avoid redundant and overly long sentences, which may sound confusing to the average reader. Examples can be found in lines 103-104, 110-114, 267-268, 301-308, 353-354, 380-382, and so on. The text as a whole was almost too long and should undergo revision to become more concise and to the point, therefore improving the reader's experience.
Author Response
Dear Reviewer 4
I have responded to all your points and included them into the manuscript (see yellow marks).
The author provides a long and thorough review of the role of the extracellular matrix in cancer development and survival. They also explain in detail how mechanosensing and matrix stiffness are related to chemotherapy resistance, tumor progression, and differentiation. The references are updated and follow the state of the art from the field. Some suggestions to improve text quality are described below:
- Double-spacing in line 62.
Answer: Thank you. Done.
- Text is found duplicated in line 134 - "... on the one hand, but on the other hand, ...".
Answer: Thank you. Done. I have made two sentences out of it.
- The paragraph ranging from lines 182 to 190 requires at least one citation.
Answer: I have added four citations. Thank you for this advice.
- Lines 208, 210, 225, and 226: please substitute the hyphens (-) with Em Dashes ( – ).
Answer: Thank you. Done.
- The design of Figure 3B is too small and crowded. I would suggest increasing the figure size to better display both the text and the diagram.
Answer: Thank you. I have followed your instructions and enlarged the fond sizes and altered the colors for increased readability. I also enlarged the total figure size.
- Figure legends for Figures 3C, 4, 6, and 12 should be expanded to include a more detailed description.
Answer: Thank you. Done.
- Line 563: Following the MISEV2018 guidelines (Théry et al, 2019 - PMID: 30637094), researchers should refrain from using the term "exosomes" unless the endocytic pathway of those extracellular vesicles is proven. If the original paper does not verify this biogenesis, authors should avoid this term and apply the broader "extracellular vesicles (EV)" terminology.
Answer: Thank you. Done.
- Lines 691-692, 851-852, 873-874, 888: check paragraph formatting.
Answer: Thank you. Done.
English Language:
Although the science is sound, I suggest an extensive review of the English language to avoid redundant and overly long sentences, which may sound confusing to the average reader. Examples can be found in lines 103-104, 110-114, 267-268, 301-308, 353-354, 380-382, and so on. The text as a whole was almost too long and should undergo revision to become more concise and to the point, therefore improving the reader's experience.
Answer: I have improved the given examples and checked the entire manuscript. Whenever possible, I have improved the text (see yellow marks).
It is difficult to meet the different requirements of several reviewers, another review even suggested adding more sections.
In my review article, I tried to cover the topic comprehensively in order to arouse the interest of many different readers, which made the article quite long. I have taken on board the reviewers' suggestions to organize and restructure the article more clearly and put a table of contents at the beginning. This makes it easier for readers with a wide range of interests to find the aspects of my work that are important to them and, if necessary, skip the points that are not so important to them or read them only briefly.
I hope this compromise solution meets with your approval.
Best regards
Claudia Tanja Mierke

Round 2
Reviewer 1 Report
Comments and Suggestions for Authors
The authors answered my questions and the quality of the manuscript has been improved greatly.
Reviewer 2 Report
Comments and Suggestions for Authors
No further comments.